# N-WASP-dependent branched actin polymerization attenuates B-cell receptor signaling by increasing the molecular density of receptor clusters

Anshuman Bhanja[1†], Margaret K Seeley-Fallen[1], Michelle Lazzaro[1‡], Arpita Upadhyaya[2,3,4]*, Wenxia Song[1]*

[1]Department of Cell Biology and Molecular Genetics, University of Maryland, College Park, United States; [2]Biophysics Program, University of Maryland, College Park, United States; [3]Department of Physics, University of Maryland, College Park, United States; [4]Institute for Physical Science and Technology, University of Maryland, College Park, United States

*For correspondence:
arpitau@umd.edu (AU);
wenxsong@umd.edu (WS)

Present address: †Department of Pharmacology, University of California, San Diego, United States; ‡AstraZeneca, Gaithersburg, United States

Competing interest: The authors declare that no competing interests exist.

**Abstract** Antigen-induced B-cell receptor (BCR) signaling is critical for initiating and regulating B-cell activation. The actin cytoskeleton plays essential roles in BCR signaling. Upon encountering cell-surface antigens, actin-driven B-cell spreading amplifies signaling, while B-cell contraction following spreading leads to signal attenuation. However, the mechanism by which actin dynamics switch BCR signaling from amplification to attenuation is unknown. Here, we show that Arp2/3-mediated branched actin polymerization is required for mouse splenic B-cell contraction. Contracting B-cells generate centripetally moving actin foci from lamellipodial F-actin networks in the plasma membrane region contacting antigen-presenting surfaces. Actin polymerization driven by N-WASP, but not WASP, initiates these actin foci and facilitates non-muscle myosin II recruitment to the contact zone, creating actomyosin ring-like structures. B-cell contraction increases BCR molecular density in individual clusters, leading to decreased BCR phosphorylation. Increased BCR molecular density reduced levels of the stimulatory kinase Syk, the inhibitory phosphatase SHIP-1, and their phosphorylated forms in individual BCR clusters. These results suggest that N-WASP-activated Arp2/3, coordinating with myosin, generates centripetally moving foci and contractile actomyosin ring-like structures from lamellipodial networks, enabling contraction. B-cell contraction attenuates BCR signaling by pushing out both stimulatory kinases and inhibitory phosphatases from BCR clusters, providing novel insights into actin-facilitated signal attenuation.

## eLife assessment

This is an **important** study highlighting a distinct role of WASP dependent actin foci in B cell antigen receptor signalling. The evidence supporting the conclusions is **compelling**. The proposal of higher molecular density in B cell receptor clustering leading to kinase exclusion and attenuated signalling is provocative as it contrasts with models for other antigen receptors.

## Introduction

B-cell-mediated antibody responses are essential for eliminating invading pathogens. BCRs expressed on the B-cell surface detect the presence of cognate antigens. The binding of antigens to the BCR leads to the activation of signaling cascades (*Reth and Wienands, 1997*; *Dal Porto et al., 2004*;

*Kwak et al., 2019*), which induce transcriptional programs that prepare B-cells for proliferation and differentiation (*Kurosaki et al., 2010*; *Shlomchik et al., 2019*; *Wang et al., 2020*). BCR signaling also induces rapid antigen internalization, processing, and presentation for T-cell recognition, which provides the second signal required for B-cell clonal expansion and differentiation to high-affinity antibody-secreting cells and memory B-cells (*Song et al., 1995*; *Gitlin et al., 2014*). BCR signaling is tightly regulated by various external and internal factors to activate antibody responses that are specific and also qualitatively and quantitatively matched to the encountered antigen.

Antigen-induced receptor reorganization activates BCRs at the B-cell surface. Antigen binding leads to BCR clustering at lipid rafts, which enables raft-resident kinases, such as the Src kinases Lyn and Fyn, to phosphorylate immunoreceptor tyrosine-based activation motifs (ITAMs) in the cytoplasmic domains of the BCR signaling subunit CD79a/b heterodimer (*Reth, 1994*; *Pierce, 2002*; *Sohn et al., 2008*). Doubly phosphorylated ITAMs recruit and activate spleen tyrosine kinase (Syk), which in turn activates multiple downstream signaling pathways, including phospholipase Cγ2 (PLCγ2), Bruton's tyrosine kinase (Btk), Ras-GTPase, and phosphatidylinositol-3 kinase (PI3K), initiating signaling cascades (*Kurosaki, 2000*; *Dal Porto et al., 2004*; *Tanaka and Baba, 2020*). Signaling activation also induces the recruitment and activation of inhibitory phosphatases, such as SH2-containing tyrosine phosphatase-1 (SHP-1) and phosphatidylinositol-5 phosphatase-1 (SHIP-1), to BCR signaling complexes (*Brauweiler et al., 2000*; *Gross et al., 2009*; *Franks and Cambier, 2018*). SHIP-1 and SHP-1 negatively regulate BCR signaling by inactivating the plasma membrane docking lipids for stimulatory kinases, such as Btk and Akt (*Aman et al., 1998*; *Bolland et al., 1998*) and dephosphorylating the BCR and its downstream signaling molecules (*Mizuno et al., 2000*; *Adachi et al., 2001*), respectively. The interplay between the stimulatory kinases and the inhibitory phosphatases controls the balance of antibody responses against pathogens and self. Deficiencies in stimulatory kinases by mutations, such as Btk, cause X-linked agammaglobulinemia (XLA) (*Kinnon et al., 1993*), while deficiencies in the inhibitory phosphatases SHP-1 or SHIP-1 result in autoimmune diseases (*Pao et al., 2007*; *Leung et al., 2013*). However, the mechanisms by which BCR signaling attenuation is initiated and regulated remain elusive.

B-cells encounter both soluble and membrane-associated antigens in vivo. Membrane-associated antigens include antigens on antigen-presenting cells, like follicular dendritic cells in B-cell follicles or germinal centers, and pathogenic cells, like bacteria, parasites, and cancer cells (*Batista and Harwood, 2009*; *Depoil et al., 2009*; *Gonzalez et al., 2009*; *Cyster, 2010*). The binding of multivalent soluble antigen and membrane-associated antigen with any valency induces dynamic reorganization of surface BCRs into microclusters, triggering BCR signaling. Subsequently, surface BCRs continue moving to the plasma membrane region contacting antigen-presenting surfaces or to one B-cell pole in the case of soluble antigen, which leads to the growth and merger of BCR microclusters and the formation of immunological synapses (*Carrasco et al., 2004*; *Harwood and Batista, 2010*) or supra-molecular activation complexes (*Unanue and Karnovsky, 1973*; *Schreiner and Unanue, 1977*; *Tolar et al., 2009b*; *Harwood and Batista, 2010*).

Multiple mechanisms have been proposed for signaling initiation by antigen-induced surface BCR reorganization. The binding of surface BCRs to membrane-associated antigen has been shown to induce a conformational change in the BCR extracellular domain, exposing a proximal membrane region of membrane IgM that promotes receptor clustering (*Tolar et al., 2009a*; *Shen et al., 2019*). Antigen-binding also changes the conformation of the BCR cytoplasmic domains from a closed to an open form, facilitating the recruitment of signaling molecules (*Tolar et al., 2005*). Alternatively, antigen-BCR interaction can increase the molecular spacing of BCRs in clusters, facilitating BCR interaction with signaling molecules (*Yang and Reth, 2010*; *Kläsener et al., 2014*). The inhibitory phosphatases SHP-1 and SHIP-1 are also recruited to BCR clusters (*Adachi et al., 2001*; *Seeley-Fallen et al., 2014*). However, the mechanisms underlying phosphatase recruitment and its relationship with BCR signaling activation complexes are unknown.

The actin cytoskeleton is essential for antigen-induced BCR reorganization on the B-cell surface in response to both soluble and membrane-associated antigen (*Harwood and Batista, 2011*; *Liu et al., 2013b*; *Song et al., 2013*; *Hoogeboom and Tolar, 2016*). Early signaling of the BCR triggers transient actin depolymerization via Rap GTPase and its downstream target cofilin (*Freeman et al., 2011*), which disassembles the existing cortical actin network that confines the lateral movement of surface BCRs (*Treanor et al., 2010*). Following this transient depolymerization, BCR signaling activates rapid actin

polymerization through the actin nucleation-promoting factors Wiskott-Aldrich Syndrome Protein (WASP) and neuronal-WASP (N-WASP), modulating BCR mobility and leading to F-actin accumulation at BCR-antigen interaction sites (*Liu et al., 2013a*; *Rey-Suarez et al., 2020*). Actin polymerization and treadmilling in B-cells stimulated by soluble antigens drive surface BCRs to cluster and move to one pole of the B-cell, forming a BCR cap (*Schreiner and Unanue, 1977*; *Liu et al., 2012a*). Upon interacting with membrane-associated antigens, B-cells organize actin into two dynamic structures. One treadmills outwards, driving B-cell membrane spreading and expanding the B-cell membrane region contacting the antigen-presenting surface (contact zone), which enables more BCRs to engage antigen (*Bolger-Munro et al., 2019*). The other creates retrograde flow towards the center of the B-cell contact zone, driving the centripetal movement and growth of BCR microclusters (*Liu et al., 2012b*). Together, these processes amplify BCR signaling (*Batista et al., 2010*; *Harwood and Batista, 2011*). The extent of B-cell spreading and the amount of BCR-antigen complexes gathered in the contact zone depend on BCR binding affinity and the density of antigen on the membrane (*Fleire et al., 2006*). Subsequent to spreading, B-cells contract, reducing the contact area, which drives BCR microclusters to merge into central clusters to form immunological synapses (*Fleire et al., 2006*; *Liu et al., 2013a*).

The actin cytoskeleton closely interplays with BCR signaling. We have previously shown that mouse primary B-cells with Btk deficiency (*Liu et al., 2011*) or double knockouts of the actin nucleation-promoting factors WASP and N-WASP (*Liu et al., 2013a*) fail to spread on antigen-presenting surfaces and establish stable interactions with membrane-associated antigen, drastically reducing BCR signaling. Btk can activate WASP and N-WASP through Vav, the guanine nucleotide exchange factor of Cdc42 and Rac, and phosphatidylinositol-5 kinase that generates phosphatidylinositol-4,5-biphosphate (PI4,5P$_2$) (*Sharma et al., 2009*; *Padrick and Rosen, 2010*). Surprisingly, B-cell-specific knockout of N-WASP (cNKO) but not *Wasp* germline knockout (WKO) enhances B-cell spreading, delays B-cell contraction, and inhibits the centralization of BCR clusters at the contact zone (*Liu et al., 2013a*). Consistent with these findings, Bolger-Munro et al. have shown that knockdown of the actin nucleation factor Arp2/3 downstream of WASP and N-WASP disrupts the dynamic actin reorganization induced by membrane-associated antigen required for BCR microcluster growth and merger (*Bolger-Munro et al., 2019*). Using B-cell-specific and germline knockouts, we showed that in addition to N-WASP, the actin motor non-muscle myosin IIA (NMIIA) (*Seeley-Fallen et al., 2022*) and the actin-binding adaptor protein Abp1 (*Seeley-Fallen et al., 2014*) are required for B-cell contraction and BCR central cluster formation. Finally, *Wang et al., 2022* recently showed that in the presence of the adhesion molecule ICAM on the antigen-presenting surface, B-cells form contractile actomyosin arcs, driving centripetal movement of BCR clusters in the B-cell contact zone. Together, these findings indicate a critical role of different actin networks in B-cell contraction and BCR signaling.

Our previous studies demonstrate that B-cell contraction is critical for BCR signaling attenuation. Delay or inhibition of B-cell contraction due to deficiencies in actin regulators, N-WASP, Abp1, or NMIIA, prolongs and/or increases BCR signaling by enhancing the activation of stimulatory kinases and suppressing the activation of inhibitory phosphatases, which elevates the production of autoantibody levels in mice (*Liu et al., 2013a*; *Seeley-Fallen et al., 2014*; *Seeley-Fallen et al., 2022*). N-WASP and NMIIA are not known to interact with kinases and phosphatases directly. The growth, merger, and centralization of antigen-BCR complexes at the B-cell contact zone during the B-cell contraction phase are associated with reduced BCR signaling capability (*Liu et al., 2013a*). These findings suggest that B-cell contraction is one of the mechanisms for inducing BCR signaling attenuation. However, how the actin cytoskeleton facilitates the transition of the spreading to the contraction phase and how B-cell contraction switches BCR signaling from amplification to attenuation is unknown.

This study explores the mechanisms by which the actin cytoskeleton remodels from spreading lamellipodia to contractile structures and by which B-cell contraction suppresses BCR signaling. Our results show that the contractile actomyosin structures responsible for B-cell contraction originate from branched actin at spreading lamellipodia. N-WASP/Arp2/3-mediated actin polymerization prolongs the lifetime of the lamellipodial actin structures, enables NMIIA recruitment, and drives their movement to the center of the B-cell contact zone. B-cell contraction increases the molecular density within individual BCR-antigen clusters, which promotes the disassociation of both the stimulatory kinase Syk and the inhibitory phosphatases SHIP-1 from BCR clusters, leading to signal attenuation. Our study reveals a new mechanism underlying BCR signal downregulation.

## Results

### Arp2/3, activated by N-WASP but not WASP, is required for B-cell contraction

Arp2/3-generated branched F-actin is known to drive lamellipodial expansion for B-cell spreading (*Bolger-Munro et al., 2019*). However, whether branched F-actin is involved in the subsequent contraction phase is unknown. To address this, we perturbed the polymerization of branched F-actin using the Arp2/3 inhibitor CK-666 (50 μM) while using its inactive derivative CK-689 as a control. Pre-warmed splenic B-cells from WT C57BL/6 mice were incubated with monobiotinylated Fab' fragment of anti-mouse IgM + G attached to planar lipid bilayers (Fab'-PLB) by biotin-streptavidin interaction and imaged live at 37 °C using interference reflection microscopy (IRM). The contact area of B-cells treated with CK-689 rapidly increased upon contacting Fab'-PLB and reached a maximum level at ≥0.5 min after the initial contact (*Figure 1A*, *Figure 1—figure supplement 1*, and *Figure 1—video 1A*). Following maximal spreading, most B-cells reduced the area of their contact zone, indicating contraction (*Figure 1A and B*, *Figure 1—figure supplement 1*, and *Figure 1—video 1A*). We classified a B-cell as contracting if its contact zone area is reduced by ≥5% for at least 10 s after reaching a maximum value. Based on the average timing for B-cell maximal spreading, we treated B-cells with CK-666 at the beginning of the incubation (Time 0), before B-cell spreading initiation, or at 2 min, when all B-cells had already spread. The effectiveness of CK-666 was detected by reduced Arp2/3 staining in the contact zone (*Figure 1—figure supplement 2*). As expected, CK-666 treatment at 0 but not 2 min reduced the kinetics of B-cell spreading (*Figure 1—figure supplement 3*). Importantly, CK-666 treatment at 0 and 2 min both significantly reduced the percentage of B-cells undergoing contraction (*Figure 1A–C* and *Figure 1—video 1A*). The inhibitory effect of CK-666 on B-cell contraction, particularly CK-666 treatment after B-cell spreading, suggests that Arp2/3-mediated branched actin polymerization is required for B-cell contraction.

WASP and N-WASP are actin nucleation-promoting factors upstream of Arp2/3 that are expressed in B-cells (*Padrick and Rosen, 2010*). To determine if either or both were responsible for activating Arp2/3 for B-cell contraction, we utilized the N-WASP inhibitor wiskostatin (Wisko), and splenic B-cells from B-cell-specific *N-wasp* knockout mice (cNKO) and germline *Wasp* knockout (WKO) mice (*Westerberg et al., 2012*; *Liu et al., 2013a*). Wisko has been shown to inhibit N-WASP activation while enhancing WASP activation in B-cells (*Figure 1—figure supplement 4*; *Liu et al., 2013a*). We found that both Wisko (10 μM) (*Figure 1D and E* and *Figure 1—video 1B*) and cNKO (*Figure 1F and G* and *Figure 1—video 1C*), but not WKO (*Figure 1H and I* and *Figure 1—video 1D*), significantly reduced the percentage of B-cells undergoing contraction, compared to the vehicle, flox, or WT controls. These results suggest that N-WASP- but not WASP-activated Arp2/3 mediates branched actin polymerization for B-cell contraction.

### Arp2/3, downstream of N-WASP, generates inner F-actin foci, driving B-cell contraction

To understand how Arp2/3 drives B-cells to transition from spreading to contraction, we identified F-actin structures associated with contracting B-cells that were sensitive to CK-666 treatment. We visualized F-actin by phalloidin staining and compared F-actin organization in the contact zone of B-cells at the spreading (2 min) and contraction (4 min) phases using TIRF. While B-cells in both spreading and contraction phases exhibited phalloidin staining outlining the contact zone (*Figure 2A*, *green arrows*), only B-cells in the contracting phase showed interior phalloidin patches brighter than the phalloidin staining at the periphery of the contact zone (*Figure 2A*, *purple arrows*). These F-actin patches were organized into a ring-like structure and resided ~1 μm behind the spreading front, surrounding an F-actin-poor center (*Figure 2A*, *purple arrows*). Here, we refer to these F-actin patches as inner F-actin foci. We identified inner F-actin foci based on whether their peak fluorescence intensity (FI) was ≥twofold of the mean fluorescence intensity (MFI) of phalloidin in the no-foci area, had diameters of ≥250 nm, and were located 1 μm away from the edge of the contact zone. We found that such inner F-actin foci were detected in >60% of B-cells in the contracting phase (4 min) but only in <20% of B-cells in the spreading phase (2 min) (*Figure 2B*). CK-666 treatment at 0 min, which inhibited B-cell contraction, significantly reduced the percentage of B-cells showing inner F-actin foci at 4 min but not at 2 min (*Figure 2B*). CK-666 treatment did not affect the phalloidin staining outlining

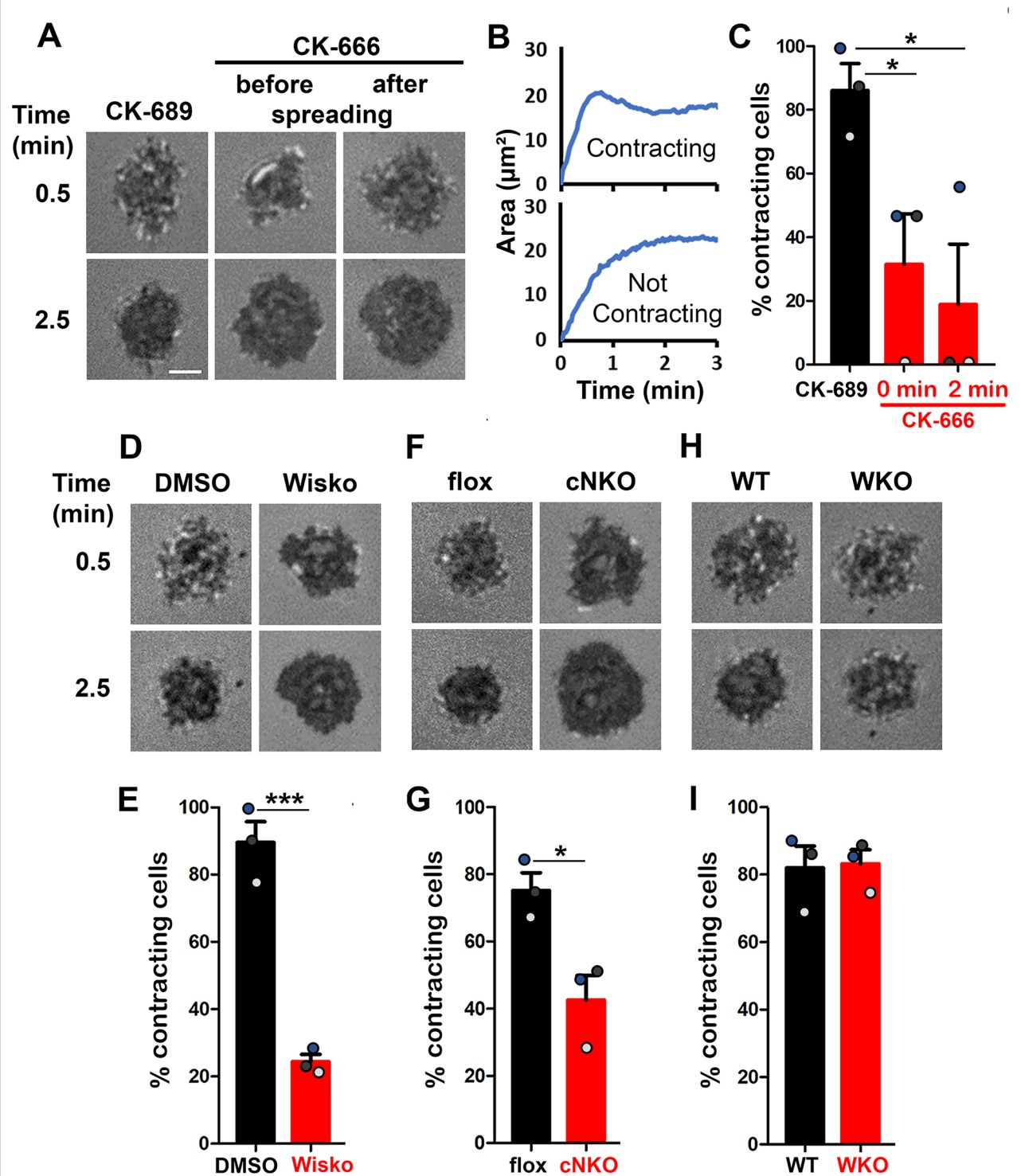

**Figure 1.** Arp2/3, activated by neuronal-Wiskott-Aldrich Syndrome Protein (N-WASP) but not WASP, is required for B-cell contraction. Splenic B-cells were incubated with planar lipid bilayers coated with monobiotinylated Fab' fragment of goat anti-mouse IgG + M (Fab'-PLB) in the absence and presence of various inhibitors and imaged live at 37 °C by interference reflection microscopy (IRM). The B-cell plasma membrane area contacting Fab'-PLB (B-cell contact zone) was measured using IRM images and custom MATLAB scripts [*Source code 1* and *Source code 2*]. (**A**) Representative IRM images of splenic B-cells from C57BL/6 mice treated with CK-689 or CK-666 (50 μM) before (0 min) and after maximal spreading (2 min). (**B**) Representative plots of the B-cell contact area versus time from one contracting cell and one non-contracting cell. (**C**) Percentages (± SEM) of B-cells that underwent contraction after treatment with CK-666 or CK-689. A B-cell was classified as contracting if its contact zone area was reduced by ≥5% for at least 10 s after reaching a maximum value. (**D**) Representative IRM images of splenic B-cells from C57BL/6 mice treated with DMSO or Wiskostatin

*Figure 1 continued on next page*

*Figure 1 continued*

(Wisko, 10 μM) 10 min before and during incubation with Fab'-PLB. (**E**) Percentages (± SEM) of B-cells that underwent contraction after treatment with Wisko or DMSO. (**F**) Representative IRM images of splenic B-cells from flox control and B-cell-specific *N-wasp* knockout (cNKO) mice. (**G**) Percentages (± SEM) of cNKO or flox control B-cells that underwent contraction. (**H**) Representative IRM images of splenic B-cells from wild-type (WT) or *Wasp* knockout mice (WKO). (**I**) Percentages (± SEM) of WKO or WT B-cells that underwent contraction. Data points in (**C**, **E**, **G** and **I**) represent three independent experiments, ~25 cells per condition per experiment, with each color representing one experiment. Scale bar, 2 μm. *p<0.05, ***p<0.001, by paired student's *t*-test. MATLAB codes were used for detecting the B-cell contact zone in time-lapse data and tracking the area occupied by it [*Source code 2*, *Source code 3*, *Source code 4*, *Source code 5*, *Source code 6*].

The online version of this article includes the following video and figure supplement(s) for figure 1:

**Figure supplement 1.** B-cells spread and contract on Fab'-coated-planar lipid bilayers.

**Figure supplement 2.** CK-666 significantly decreases Arp2/3 recruitment to the B-cell contact zone.

**Figure supplement 3.** CK-666 treatment before but not after maximal B-cell spreading decreased the spreading kinetics.

**Figure supplement 4.** Wiskostatin treatment inhibits neuronal-WASP (N-WASP) activation while enhancing WASP activation in B-cells.

**Figure 1—video 1.** Effects of CK-666, Wiskostatin, conditional neuronal-Wiskott-Aldrich Syndrom Protein (N-WASP) knockout, and WASP knockout on B-cell contraction.

https://elifesciences.org/articles/87833/figures#fig1video1

the contact zone (*Figure 2A*). Similarly, the percentage of cNKO B-cells showing inner F-actin foci was drastically reduced at 4 min but not at 2 min (*Figure 2C*). These results suggest that the formation of these inner F-actin foci is associated with B-cell contraction.

We further quantified the number of inner F-actin foci in individual B-cells from mice expressing the LifeAct-GFP transgene (*Riedl et al., 2010*; *Figure 2D, E and G*), which allowed us to monitor F-actin reorganization using live-cell imaging (*Figure 2—video 1*), or using phalloidin staining in flox control and cNKO B-cells (*Figure 2F*). Consistent with the results with phalloidin staining, CK-666 treatment at time 0 significantly reduced the number of inner F-actin foci in the contact zone (*Figure 2D* and *Figure 2-video 1A and B*). When we followed the same B-cells before and after CK-666 treatment at 2 min, many of the inner F-actin foci formed before the treatment (1 min 50 s) disappeared after the treatment (2 min 20 s), significantly reducing the number of inner F-actin foci (*Figure 2E*). Similar to the CK-666 treatment, cNKO cells significantly reduced the number of inner F-actin foci in the contact zone, compared to flox controls (*Figure 2F*). In contrast, WKO, which does not affect B-cell contraction, did not significantly change the number of inner F-actin foci, compared to WT control B-cells (*Figure 2G* and *Figure 2—video 1C and D*). Thus, N-WASP- but not WASP-activated Arp2/3 drives the formation and the maintenance of contraction-associated inner F-actin foci.

## Inner F-actin foci are derived from lamellipodial actin networks supporting the spreading membrane

We next examined the formation of contraction-associated inner F-actin foci utilizing live-cell TIRF imaging of B-cells from mice expressing LifeAct-GFP. We generated kymographs along eight lines from the center of each contact zone using time-lapse images of LifeAct-GFP (*Figure 3A and B* and *Figure 3—figure supplement 1*). Analysis of these kymographs showed that most F-actin foci were first detected closely behind lamellipodial F-actin networks. Following maximal spreading, these F-actin foci moved centripetally while increasing in intensity, becoming inner F-actin foci, in cells transitioning from spreading to contraction (*Figure 3B*, top panel, and *Figure 3—figure supplement 1*). However, such centripetally moving F-actin foci were not detected in B-cells that did not undergo contraction (*Figure 3B*, middle panel) and WKO B-cells (*Figure 3B*, bottom panel). We quantified the percentage of the eight kymographs from each cell that exhibited lamellipodia-derived inner F-actin foci and found, on average, that six out of eight kymographs from contracting cells showed lamellipodia-derived inner F-actin foci, compared to only one or two kymographs from non-contracting cells (*Figure 3C*). To examine the temporal relationship between the generation of lamellipodia-derived inner F-actin foci and contraction, we plotted the percentage of kymographs with lamellipodia-derived inner F-actin foci over time with the spreading to contracting transition time set as 0 (*Figure 3D*). We found that the percentage of kymographs showing lamellipodia-derived inner F-actin foci peaked at almost the same time when the spreading transitioned to contraction (*Figure 3D*), suggesting a close temporal relationship between the two events. Furthermore, the N-WASP inhibitor Wisko, but not

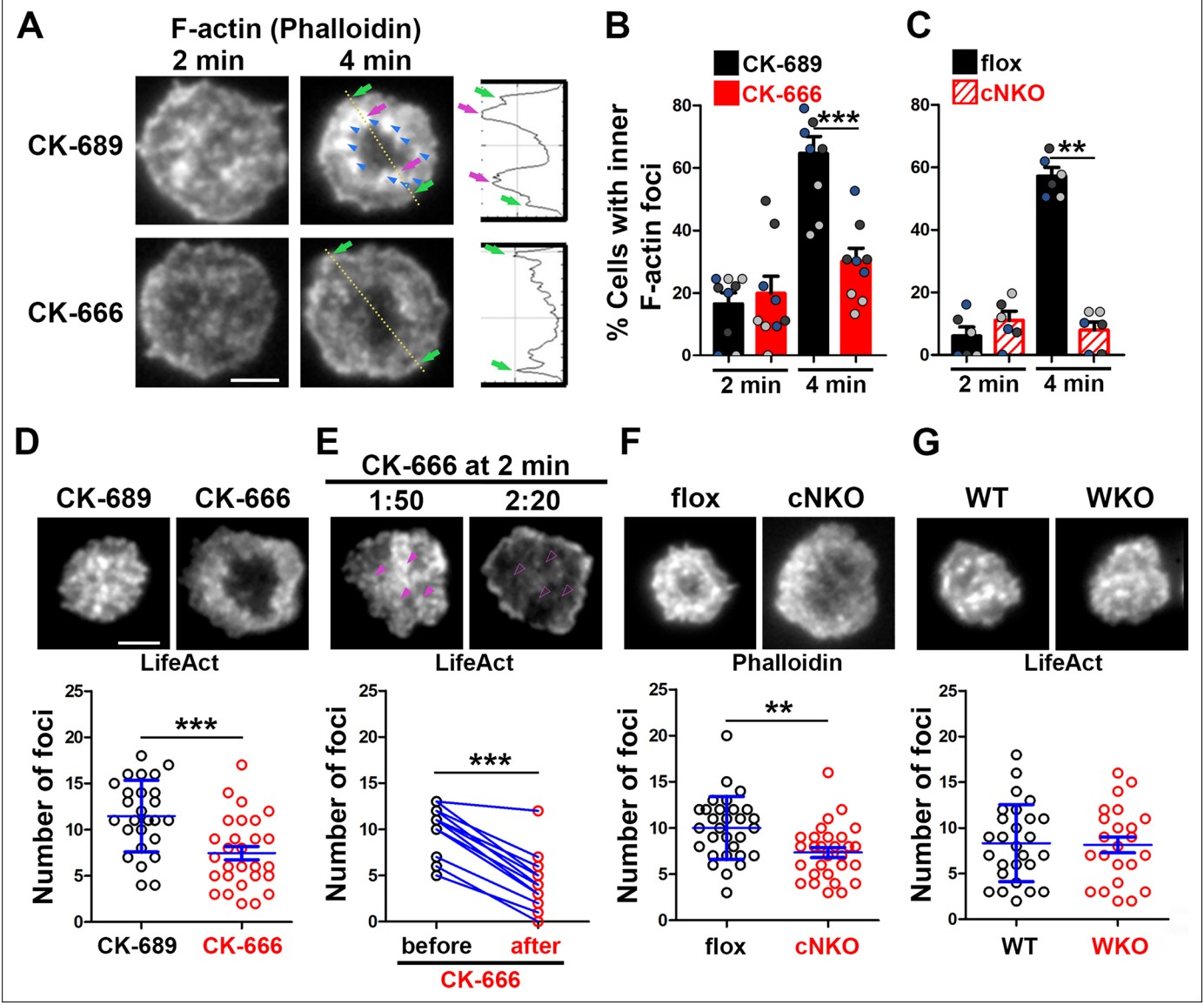

**Figure 2.** Arp2/3, downstream of N-WASP, generates inner F-actin foci, driving B-cell contraction. (**A–C**) Wild-type (WT) splenic B-cells were treated with CK-689 or CK-666 (50 µM) during incubation with Fab'-PLB (**A** and **B**), and flox control and B-cell-specific *N-wasp* knockout mice (cNKO) B-cells were incubated with Fab'-PLB at 37 °C (**C**). Cells were fixed at 2 and 4 min, permeabilized, stained for F-actin with phalloidin, and analyzed using total internal reflection fluorescence microscopy (TIRF). Shown are representative TIRF images of phalloidin staining in the contact zone of CK-689 and CK-666-treated B-cells (**A** left panels) and fluorescence intensity (FI) profiles of phalloidin staining along a line crossing cells (**A** right panels). Green arrows indicate lamellipodial F-actin, purple arrows indicate inner F-actin foci on the line, and blue arrows indicate all inner actin foci forming a ring-like structure. Percentages of cells (per image) (± SEM) with inner F-actin foci forming ring-like distribution among CK-689- versus CK-666-treated cells (**B**) and flox control versus B-cell-specific *N-wasp* knockout mice (cNKO) B-cells (**C**) before (2 min) and after (4 min) contraction were determined by visual inspection of phalloidin FI line-profiles across the B-cell contact zones. Data points in (**B**) and (**C**) represent three independent experiments, with each color representing one experiment, five images per condition per experiment, and ~15 cells per image. (**D–G**) Inner F-actin foci were identified by their diameter (≥250 nm), peak FI (≥twofold of no foci area, and location 1 µm away from the outer edge) and quantified as the number per cell using TIRF images. (**D**) Shown are representative images of splenic B-cells from LifeAct-GFP-expressing mice treated with CK-689 or CK-666 from 0 min during incubation with Fab'-PLB at 37 °C (top) and the average number (± SEM) of inner LifeAct-GFP foci per cell (bottom) at 2 min. (**E**) LifeAct-GFP B-cells were treated with CK-666 at 2 min. Shown are representative TIRF images of LifeAct-GFP in the contact zone of B-cells (top) and the average number (± SEM) of inner LifeAct-GFP foci (bottom) in the same cell 10 s before and 20 s after CK-666 treatment. Arrows indicate disappeared actin foci after CK666 treatment. (**F**) Shown are representative TIRF images of phalloidin-staining in the contact zone of flox control and cNKO B-cells after incubating with Fab'-PLB for 2 min (top) and the average number (± SEM) of inner F-actin foci per cell (bottom). (**G**) Shown are representative TIRF images of wild-type (WT) and *Wasp* germline knockout (WKO) B-cells expressing LifeAct-GFP incubated with Fab'-PLB for 2 min and the average number (± SEM) of inner

*Figure 2 continued on next page*

*Figure 2 continued*

F-actin foci per cell (bottom). Data points represent individual cells from three independent experiments with 10 (**D**, **F**, and **G**) or 6 (**E**) cells per condition per experiment. Scale bar, 2 μm. **p<0.01, ***p<0.001, by non-parametric student's *t*-test.

The online version of this article includes the following video for figure 2:

**Figure 2—video 1.** Effects of CK-666 and *Wasp* germline knockout (WKO) on F-actin foci formation.

https://elifesciences.org/articles/87833/figures#fig2video1

WKO, significantly inhibited the formation of lamellipodia-derived inner F-actin foci (*Figure 3E*). Thus, inner F-actin foci originate from branched actin-driven lamellipodia and form simultaneously with the transition of B-cell spreading to contraction.

## N-WASP-activated Arp2/3 generates inner F-actin foci by sustaining the lifetime and the centripetal movement of lamellipodial F-actin

We examined the mechanism by which N-WASP and Arp2/3 generate inner F-actin foci by measuring their relative lifetime and mobility using kymographs generated from TIRF time-lapse images of B-cells expressing LifeAct-GFP. Inner F-actin foci were identified as described above, and their tracks were manually determined (*Figure 4A*, *black dashed lines*). The time window in which an inner F-actin patch could be detected in a kymograph was measured as the relative lifetime, as actin foci could move away from the kymograph line or the TIRF evanescent field (*Figure 4A*, *right panels*). The distance each F-actin focus moved during its lifetime was used to calculate its speed (*Figure 4A*, *right panels*). Compared to CK-689-treated B-cells, CK-666 treatment at time 0 significantly reduced the relative lifetime and the centripetal speed of inner F-actin foci (*Figure 4B–D*). After B-cells were treated with CK-666 at 2 min (the average time for B-cells to reach the maximal spreading), the relative lifetime and the centripetal speed of inner F-actin foci were significantly lower than before the treatment in the same cell (*Figure 4E–G*, *line-linked dots*). Similarly, the N-WASP inhibitor Wisko significantly reduced the relative lifetime and the centripetal speed of inner F-actin foci (*Figure 4H–J*); however, WKO had no significant effect (*Figure 4K–M*). These results show that N-WASP and Arp2/3 mediated branched actin polymerization prolong the lifetime of lamellipodia-derived F-actin foci and drives them to move inward in the B-cell contact zone.

## N-WASP coordinates with NMII to generate inner actin foci and NMII ring-like structures

As non-muscle myosin II (NMII) is required for B-cell contraction (*Seeley-Fallen et al., 2022*), we examined the relationship between the formation of inner F-actin foci and the recruitment and reorganization of NMII in the contact zone using TIRF imaging of B-cells from mice expressing a GFP-NMIIA transgene. Upon interacting with Fab'-PLB, the GFP-NMIIA MFI in the contact zone of untreated WT B-cells increased rapidly in the first minute and slowly afterward (*Figure 5A and B* and *Figure 5—video 1*). Wiskostatin treatment significantly reduced the GFP-NMIIA MFI in the contact zone (*Figure 5B*) and its initial rate of increase (as determined by the slope of the GFP-NMIIA MFI increase at the 0~30 s time window) (*Figure 5C*). Kymographs generated from time-lapse TIRF images of B-cells from mice expressing GFP-NMIIA and LifeAct-RFP showed that recruited NMIIA accumulated between lamellipodia and inner F-actin foci when the foci moved centripetally away from lamellipodia (*Figure 5D*, *white arrow*, and *Figure 5—video 1*). Recruited NMIIA reorganized with inner F-actin foci to form a ring-like structure in the contact zone (*Figure 5A, D and E*, and *Figure 5—video 1*). The percentage of B-cells with NMIIA ring-like structures, visualized by immunostaining, increased over time as more B-cells underwent contraction (*Figure 5E and F*). Compared to flox controls, the percentage of cNKO B-cells with NMIIA ring-like structures was significantly decreased (*Figure 5E and F*). Wiskostatin treatment also reduced NMIIA recruitment and ring-like structure formation (*Figure 5—video 1*). Surprisingly, the percentage of WKO B-cells with NMIIA ring-like structure was higher than that of flox control B-cells (*Figure 5E and F*). Thus, N-WASP and Arp2/3 mediated branched actin polymerization promotes the recruitment and the organization of NMII ring-like structures.

We next tested whether recruited NMII contributed to the formation of inner actin foci by inhibiting its motor activity using blebbistatin, which is known to impede B-cell contraction (*Seeley-Fallen et al., 2022*). Inner F-actin foci in the B-cell contact zone were identified by phalloidin staining, as described

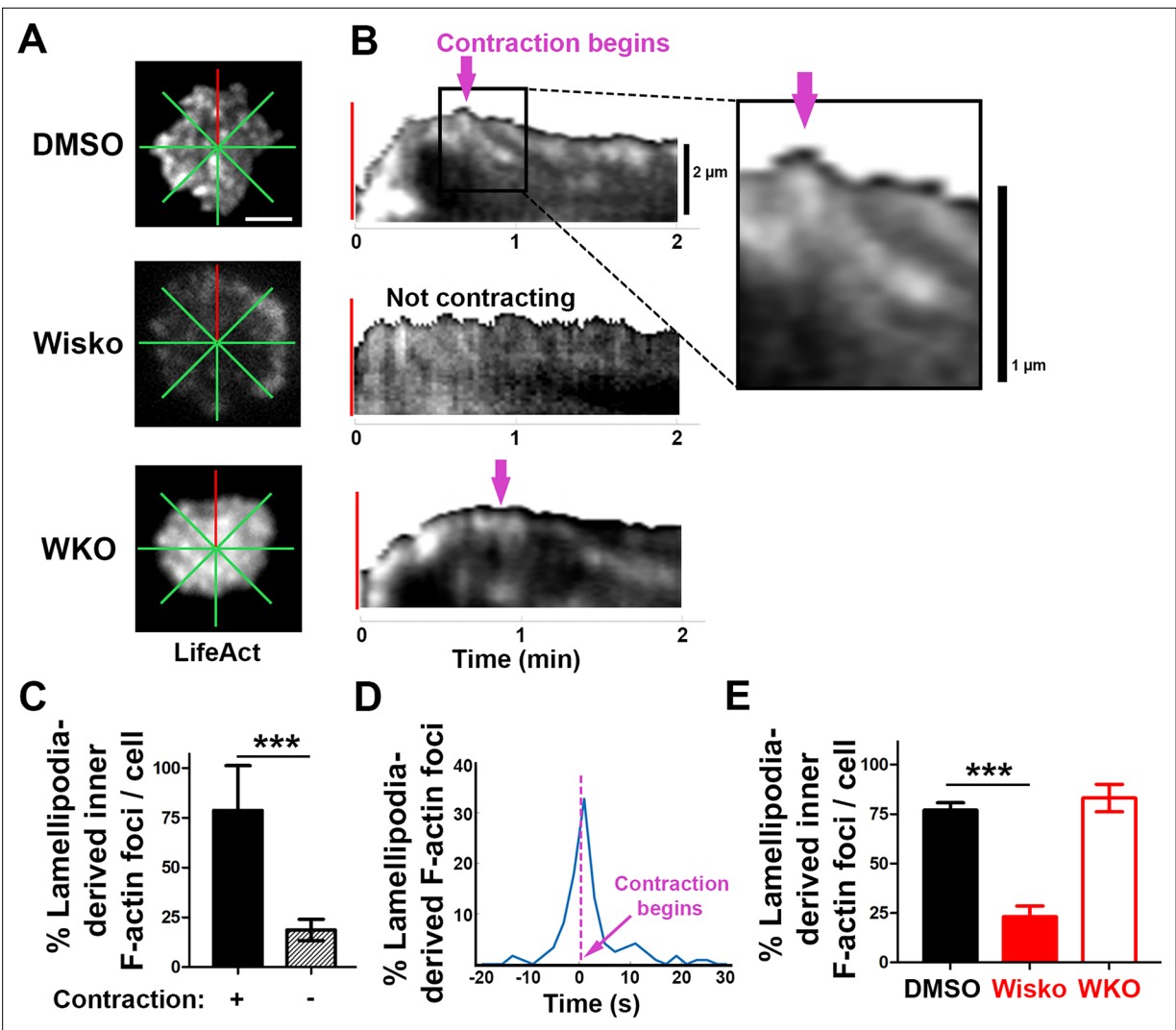

**Figure 3.** Inner F-actin foci are originated from lamellipodia behind the spreading membrane. Mouse splenic B-cells from LifeAct-GFP transgenic mice were treated with DMSO or Wisko (10 µM), imaged live using total internal reflection fluorescence microscopy (TIRF) during incubation with Fab'-PLB at 37 °C, and analyzed using kymographs generated by Fiji ImageJ. (**A**) One frame of TIRF time-lapse images of LifeAct-GFP in the contact zone of B-cells treated with DMSO or Wisko and a *Wasp* germline knockout (WKO) B-cell. Lines indicate eight kymographs that were randomly generated from each cell. (**B**) Representative kymographs were generated from TIRF time-lapse images of LifeAct-GFP at the red line in (**A**). Top panel, a contracting cell. Arrows indicate the start of contraction with inner F-actin foci originating from lamellipodia. Middle panel, a non-contracting cell. Bottom panel, a WKO cell. Lamellipodia-derived inner F-actin foci were identified by their LifeAct-GFP FI ≥twofold of their nearby region, migrating out of the lamellipodial F-actin toward the center of the contact zone, and trackable for >8 s. (**C**) Percentages (± SEM) of kymographs showing inner F-actin foci originating from lamellipodia per cell that did and did not undergo contraction. Data were generated from three independent experiments with ~10 cells per condition per experiment. (**D**) A histogram of inner F-actin foci emerging (expressed as percentages of the total events, blue line) over time relative to the time of B-cell contraction (defined as 0 s, indicated by a purple dash line and arrow). Data were generated from five independent experiments with ~9 cells per condition per experiment. (**E**) Percentage (± SEM) of inner F-actin foci originated from lamellipodia observed in eight randomly positioned kymographs of each DMSO- or Wisko-treated wild-type (WT) or untreated WKO B-cell. Data were generated from three independent experiments with ~10 cells per condition per experiment. Scale bars, 2 µm. ***p<0.001, by non-parametric student's *t*-test. MATLAB codes were used for generating kymographs from time-lapse data [*Source code 7*].

The online version of this article includes the following figure supplement(s) for figure 3:

**Figure supplement 1.** Emerging of Inner F-actin foci from lamellipodia.

in *Figure 2*. Treatment with blebbistatin significantly reduced the percentages of WT splenic B-cells exhibiting inner F-actin foci ring-like structures (*Figure 5G and H*). This result suggests that while NMII is recruited with the help of N-WASP and Arp2/3 mediated actin polymerization, its motor activity is critical for the maturation of F-actin structures associated with B-cell contraction.

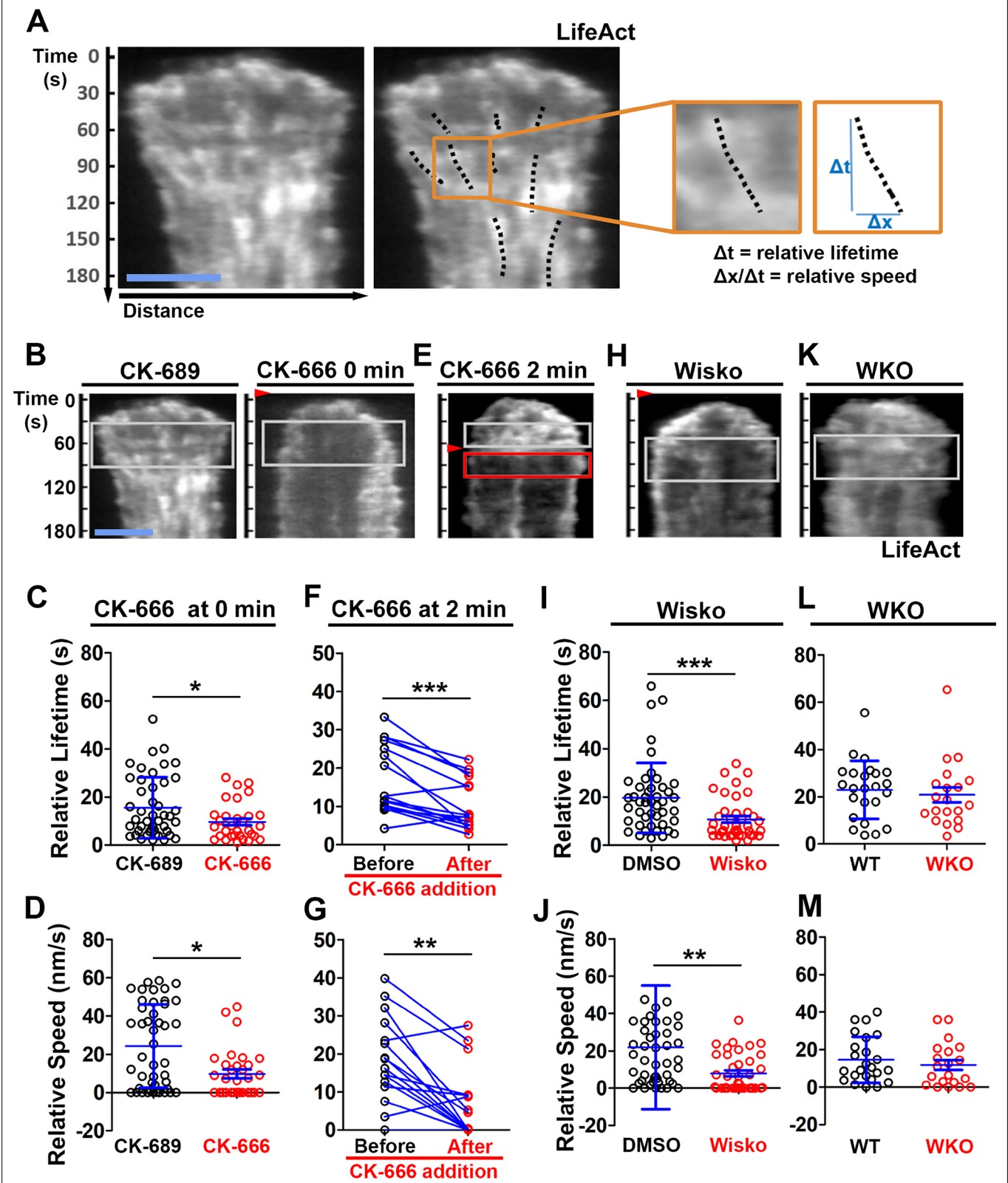

**Figure 4.** N-WASP-activated Arp2/3 sustains the lifetime and the centripetal movement of inner F-actin foci. LifeAct-GFP-expressing B-cells were incubated with or without various inhibitors and imaged live by total internal reflection fluorescence microscopy (TIRF) during interaction with Fab'-PLB at 37 °C. Three kymographs were generated for each cell using time-lapse images and positioned to track as many inner actin foci as possible. (**A**) A representative kymograph from TIRF time-lapse images of a DMSO-treated B-cell (Left panels). Inner F-actin foci were identified as described in

*Figure 4 continued on next page*

*Figure 4 continued*

*Figure 3*, and those that emerged during the 60 s window after maximal spreading (white rectangles in B) and can be tracked for ≥4 s (dashed lines) in individual kymographs were analyzed. Relative lifetimes of inner F-actin foci were measured using the duration each focus could be detected in a kymograph. The relative distances traveled by the foci were measured using the displacement of each focus in a kymograph. Relative speed was calculated for each inner F-actin focus by dividing its relative distance by its relative lifetime (right panels). (**B–D**) B-cells were treated with CK-689 or CK-666 (50 μM) from the beginning of the incubation with Fab'-PLB (0 min). Shown are representative kymographs (**B**), relative lifetimes (**C**), and relative speed (**D**) of inner F-actin foci in CK-689- versus CK-666-treated B-cells. (**E–G**) B-cells were treated with CK-666 at maximal spreading (2 min). Shown are a representative kymograph of a CK-666-treated cell (**E**), relative lifetimes (**F**), and relative speeds (**G**) of inner F-actin foci in 30 s windows before the inhibition and 10 s after the inhibition in the same cells (linked by blue lines). (**H–J**) B-cells were treated with DMSO or Wisko (10 μM) 10 min before and during interaction with Fab'-PLB. Shown is a representative kymograph of a Wisko-treated B-cell (**H**), relative lifetime (**I**), and relative speed (**J**) of inner F-actin foci in DMSO versus Wisko-treated B-cells. (**K–M**) LifeAct-GFP-expressing wild-type (WT) and *Wasp* germline knockout (WKO) B-cells were incubated with Fab'-PLB. Shown are a representative kymograph of a WKO B-cell (**K**), relative lifetime (**L**), and relative speed (**M**) of WT versus WKO B-cells. Data points represent the averaged values from inner F-actin foci in individual cells, with three kymographs per cell and ~12 cells per condition per experiment from three independent experiments. Scale bar, 2 μm. *$p<0.05$, **$p<0.01$, ***$p<0.001$, by non-parametric and paired student's *t*-test.

## B-cell contraction increases the BCR molecular density in individual clusters

To understand how B-cell contraction promotes BCR signaling attenuation, we examined the impact of B-cell contraction on the properties of BCR clusters. We first measured the MFI of AF546-Fab' attached to PLB gathered by B-cells into the contact zone as an indication of the overall BCR molecular density. The clustering of AF546-Fab' on PLB by B-cell binding reflects surface BCR clustering, as B-cell binding to transferrin (Tf)-tethered PLB does not cause surface BCRs to cluster and be phosphorylated (*Figure 6—figure supplement 1 Liu et al., 2011; Liu et al., 2013a*). The MFI of AF546-Fab' in the B-cell contact zone increased over time. Treatment with CK-666, Wisko, or cNKO all reduced the AF546-Fab' MFI, particularly during the time window of B-cell contraction in controls (*Figure 6A–G*, purple rectangles, and *Figure 6—video 1*). Notably, the rates of increase in AF546-Fab' MFI, calculated from the slopes of AF546-Fab' MFI versus time plots in individual cells, were significantly higher during B-cell contraction than before B-cell contraction in control cells and conditions (*Figure 6D–K*). Significantly, inhibiting B-cell contraction by CK-666 treatment at 0 (*Figure 6A, D and H*, and *Figure 6-video 1A and B*) or 2 min (*Figure 6A, E and I*, and *Figure 6-video 1A and B*), Wisko (*Figure 6B, F and J*, and *Figure 6-video 1C and D*), and cNKO (*Figure 6C, G and K*, and *Figure 6-video 1E and F*) reduced the increases in AF546-Fab' accumulation rates. We further examined the peak FI of AF546-Fab' in individual microclusters as a measure of the BCR molecular density in individual clusters. AF546-Fab' clusters were identified based on their diameters ≥250 nm, peak FI ≥1.1 fold outside the B-cell contact zone, and trackable for ≥20 s (*Figure 6L* and *Figure 6—figure supplement 2*). AF546-Fab' microclusters could not be identified during the early stage of B-cell spreading. Time-lapse imaging by TIRF enabled us to measure the rate of increase in AF546-Fab' peak FI in individual clusters (*Figure 6M* and *Figure 6—figure supplement 2*). Consistent with our observation of AF546-Fab' MFI increase in the contact zone, the peak FI of individual clusters increased at a faster rate during contraction than after contraction (when the contact area no longer decreased) (*Figure 6N*). Furthermore, CK-666 (*Figure 6O*), Wisko (*Figure 6P*), and cNKO (*Figure 6Q*) all significantly reduced the rate of increase in AF546-Fab' peak FI in individual clusters. These results show that B-cell contraction significantly increases the molecular density of BCRs in BCR clusters.

## Increased BCR molecular density by B-cell contraction reduces BCR phosphorylation levels in individual microclusters

To examine how BCR molecular density influenced BCR signaling capability, we immunostained B-cells interacting with AF546-Fab'-PLB for 1, 3, 5, and 7 min, for phosphorylated CD79a (pCD79a, Y182) and performed IRM and TIRF imaging (*Figure 7A and H*). We analyzed equal numbers of AF546-Fab' clusters selected randomly, in the contact zone of B-cells interacting with Fab'-PLB for 1, 3, 5, and 7 min, using a gradient threshold of their AF546-Fab' MFI, 1.1–4.1 fold of the background, and with a diameter ≥250 nm (*Figure 6—figure supplement 2*). We determined the MFI of pCD79a and AF546-Fab' in individual clusters and plotted the MFI ratio of pCD79a relative to AF546-Fab' (reflecting the relative level of BCR phosphorylation) versus the AF546-Fab' peak FI (reflecting BCR molecular density) in individual clusters (*Figure 7B and C*). The dot plots show that individual AF546-Fab' clusters with

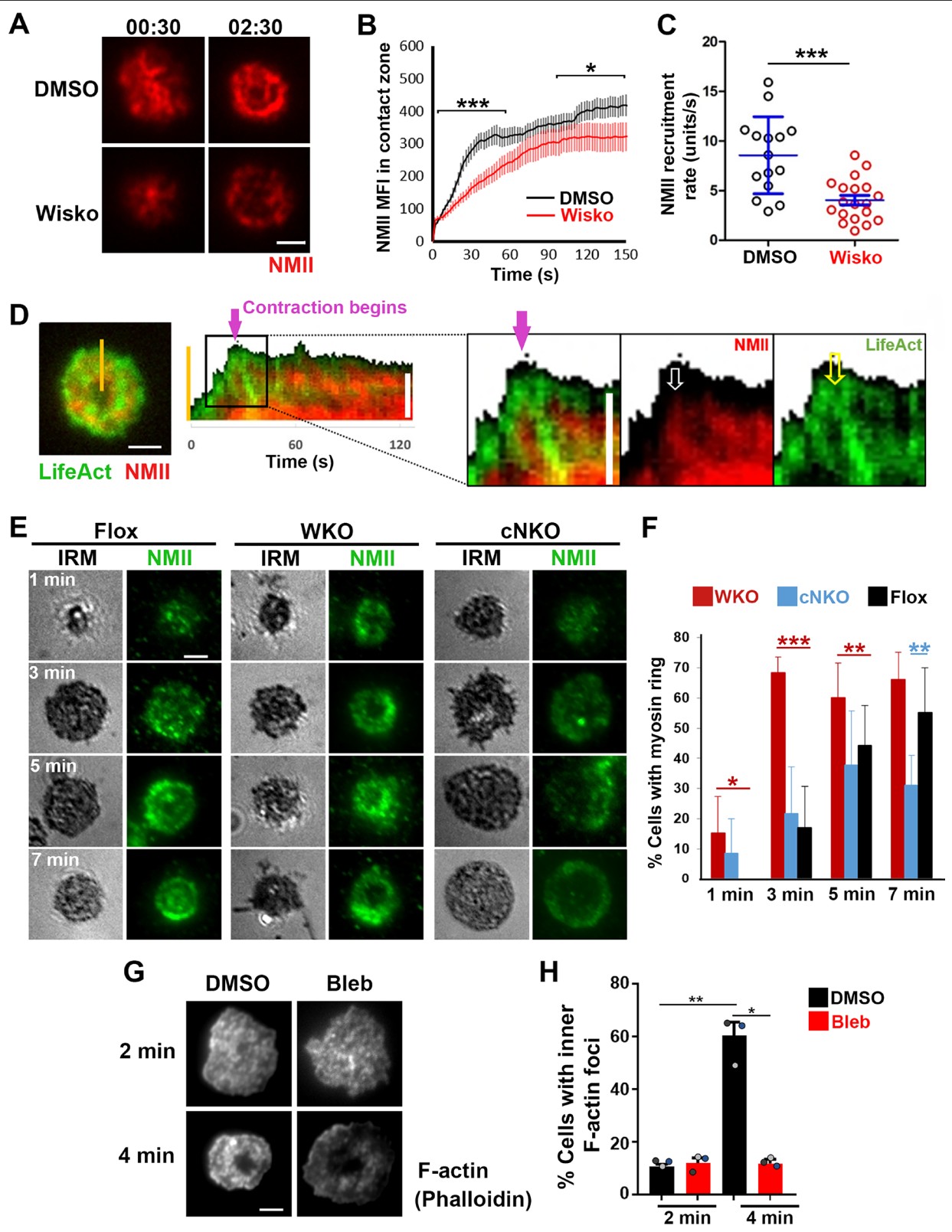

**Figure 5.** Neuronal-WASP (N-WASP) facilitates non-muscle myosin II (NMII) recruitment and ring-like structure formation, and NMII promotes inner F-actin foci ring maturation. (**A–C**) B-cells from mice expressing the GFP fusion of non-muscle myosin IIA (GFP-NMIIA) transgene were treated with DMSO or Wisko (10 μM) 10 min before and during incubation with Fab'-PLB. The B-cell contact zones were imaged live using total internal reflection fluorescence microscopy (TIRF). Shown are representative TIRF images of DMSO- and Wisko-treated B-cells at 30 s (during spreading) and 2 min 30 s

*Figure 5 continued on next page*

*Figure 5 continued*

(after maximal spreading) post landing (**A** Scale bars, 2 μm), the averaged GFP-NMIIA MFI (± SEM) (**B**), and the initial rates of increasing (± SEM) of GFP-NMIIA in the contact zone (the slope of the initial GFP-NMIIA MFI versus time curves of individual cells) (**C**). Data points represent individual cells from three independent experiments with ~6 cells per condition per experiment. *p<0.05, ***p<0.001, by Kolmogorov-Smirnov test (**B**) or non-parametric student's *t*-test (**C**). (**D**) Primary B-cells from mice expressing both GFP-NMIIA and LifeAct-RFP transgenes were incubated with Fab′-PLB at 37 °C and imaged live by TIRF. Shown are a representative TIRF image of a cell and a kymograph generated from time-lapse TIRF images at the yellow line. The purple arrow indicates the starting point of contraction, the white arrow GFP-NMIIA recruitment proximal to the spreading membrane, and the yellow arrow an F-actin (LifeAct-RFP) focus originating at the lamellipodia and moving away from the spreading membrane. (**E, F**) Primary B-cells from flox control, *Wasp* germline knockout (WKO), and B-cell-specific *N-wasp* knockout mice (cNKO) mice were incubated with Fab′-PLB for indicated times. Cells were fixed, permeabilized, stained for non-muscle myosin II (NMII) light chain, and imaged by interference reflection microscopy (IRM) and TIRF. Shown are representative IRM and TIRF images (**E**) and percentages (± SD) of B-cells with the NMII ring-like structure in individual images (**F**), identified by visual inspection. The data were generated from three independent experiments with five images per condition per experiment. Scale bars, 2 μm. *p<0.05, **p<0.01, ***p<0.001, by non-parametric student's *t*-test. (**G, H**) Wild-type (WT) splenic B-cells were treated with DMSO or Blebbistatin (Bleb, 50 μM) 20 min before and during incubation with Fab′-PLB at 37 °C. Cells were fixed at 2 and 4 min, permeabilized, stained with phalloidin, and imaged by TIRF. Shown are representative TIRF images of the B-cell contact zone (**G**) and percentages of cells (± SEM) with inner F-actin foci forming ring-like distribution (**H**), determined as described in *Figure 2*. Data were generated from three independent experiments with ~50 cells per condition per experiment with different color dots representing individual experiments. Scale bar, 2 μm. *p<0.05, **p<0.01 by paired student's *t*-test. MATLAB codes were used for detecting the B-cell contact zone and quantifying NMII FI [*Source code 1*, *Source code 8* and *Source code 9*].

The online version of this article includes the following video for figure 5:

**Figure 5—video 1.** Wiskostatin treatment inhibits non-muscle myosin II (NMII) ring-like structure formation.
https://elifesciences.org/articles/87833/figures#fig5video1

relatively low peak FI displayed increasing MFI ratios of pCD79a to AF546-Fab′ as the AF546-Fab′ peak FI rose in flox control B-cells (*Figure 7B*). Past a certain level of AF546-Fab′ peak FI, the MFI ratios of pCD79a to AF546-Fab′ decreased as the AF546-Fab′ peak FI in individual clusters further increased (*Figure 7A and B*). Inhibition of B-cell contraction by cNKO reduced the AF546-Fab′ peak FI in individual clusters, maintaining it within a relatively low range, where the MFI ratios of pCD79a to AF546-Fab′ increased with the AF546-Fab′ peak FI (*Figure 7C and D*). Additionally, the average pCD79a to Fab′ MFI ratios were much higher in cNKO B-cells than flox control B-cells, when comparing AF546-Fab′ clusters with the same range of peak FI (*Figure 7B–D*). During the contraction stage (5 and 7 min), flox control B-cells exhibited increases in AF546-Fab′ MFI (*Figure 7E*) but decreases in pCD79a MFI (*Figure 7F*) and the MFI ratios of pCD79a relative to AF54-Fab′ (*Figure 7G*) in individual clusters, compared to the spreading stage (1 and 3 min). In contrast, non-contracting cNKO B-cells only slightly increased AF546-Fab′ MFI (*Figure 7E*) but significantly increased pCD79a MFI (*Figure 7F*), and the pCD79a to AF546-Fab′ MFI ratios (*Figure 7G*) in individual clusters at the contraction stage compared to the spreading stage. Consequently, individual clusters in non-contracting cNKO B-cells had significantly lower AF546-Fab′ MFI but significantly higher pCD79a MFI and pCD79a to AF546-Fab′ FIRs than contracting flox control B-cells at 5 and 7 min but not at 1 and 3 min (*Figure 7E–G*). Similarly, inhibiting B-cell contraction by treatment with the Arp2/3 inhibitor CK-666 at 2 min post-stimulation reduced AF546-Fab′ MFI (*Figure 7I*) but increased pCD79a MFI (*Figure 7J*) and the MFI ratios of pCD79a relative to AF546-Fab′ (*Figure 7K*) in individual clusters. These data suggest that increases in BCR molecular density in BCR clusters during B-cell contraction inhibit BCR phosphorylation.

## Increased BCR molecular density by B-cell contraction promotes the disassociation of the stimulatory kinase Syk from BCR microclusters

Increased BCR molecular density may promote signaling attenuation by inducing the disassociation and/or dephosphorylation of stimulatory kinases from and at BCR clusters. To test this hypothesis, we analyzed the relative amounts of Syk, a major stimulatory kinase in the BCR signaling pathway, and its phosphorylated form pSyk (Y519/520) in individual BCR clusters in relation to the molecular density of BCRs. Splenic B-cells were incubated with Fab′-PLB for 3 and 7 min (when most cells were at the spreading and contraction phase, respectively), fixed, permeabilized, stained for Syk (*Figure 8A*) or pSyk (*Figure 8E*), and imaged using TIRF. We measured MFI ratios of Syk relative to AF546-Fab′ in individual BCR clusters to reflect the relative amount of Syk associated with individual BCR clusters and analyzed their relationship with AF546-Fab′ peak FI (reflecting the molecular density within BCR clusters). AF546-Fab′ clusters were detected and analyzed as described above. In flox control B-cells, the highest fractions of BCR clusters had an AF546-Fab′ peak FI at the 150~200 range (*Figure 8A*,

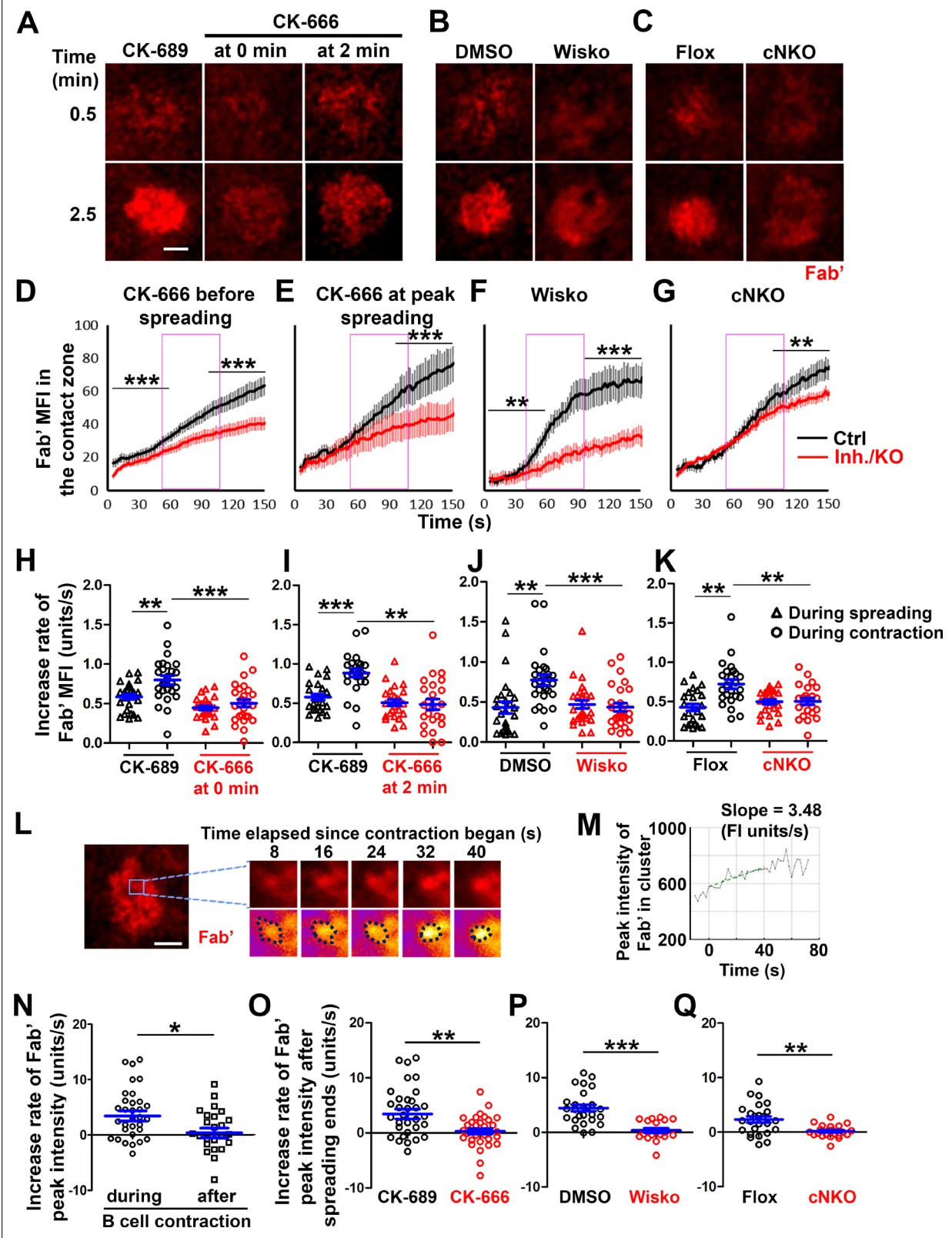

**Figure 6.** B-cell contraction increases the molecular density within B-cell receptor (BCR) clusters. Primary B-cells from wild-type (WT) mice were treated with CK-689 or CK-666 from the beginning of the incubation with AF546-Fab'-PLB (0 min) or at the maximal B-cell spreading (2 min) (**A, D, E, H, I, N, O**). WT B-cells were treated with DMSO or Wisko (10 µM) 10 min before and during the incubation with AF546-Fab'-PLB (**B, F, J, P**). B-cells from flox control and B-cell-specific *N-wasp* knockout mice (cNKO) mice were incubated with AF546-Fab'-PLB (**C, G, K, Q**). The B-cell contact zones were imaged

*Figure 6 continued on next page*

*Figure 6 continued*

live by total internal reflection fluorescence microscopy (TIRF). (**A–C**) Representative time-lapse images at 30 s (during B-cell spreading) and 2 min 30 s (after maximal spreading) after cell landed. Scale bars, 2 µm. (**D–G**) The mean fluorescence intensity (MFI) of AF546-Fab' in the contact zone was plotted over time. Purple rectangles indicate the contraction phase based on the changes in the average B-cell contact area over time. (**H–K**) Rates of AF546-Fab' MFI increases in the B-cell contact zone before and during contraction were determined by the slope of AF546-Fab' MFI versus time plots. Data points represent individual cells and were generated from three independent experiments with 6–12 cells per experiment. *p<0.05, **p<0.01, ***p<0.001, by Kolmogorov-Smirnov test (**D–G**) or paired student's t-test (**H–K**). (**L**) A representative frame from a time-lapse of a CK-689-treated B-cell (left) shows AF546-Fab' clusters 30 s after contraction began, and enlarged time-lapse images (right) show a single AF546-Fab' cluster over a 40 s time window after contraction began. AF546 clusters were identified using the criteria: ≥250 nm in diameter, ≥1.1 fold of fluorescence intensity (FI) outside the B-cell contact zone, and trackable for ≥20 s. Scale bars, 2 µm. (**M**) The peak FI of an AF546-Fab' cluster was measured over time, and the increasing rate of AF546-Fab' peak FI of this cluster was determined by the slope of the plot. (**N–Q**) The rates (± SEM) of increase in AF546-Fab' peak FI in individual clusters were compared between during and after contraction (**N**), between B-cells treated with CK-689 and CK-666 from 0 min (**O**), between DMSO- and Wisko-treated B-cells (**P**), and between flox control and cNKO B-cells (**Q**) after B-cells reached maximal spreading. Data points represent individual cells, the averaged slopes of clusters detected in one B-cell, from three independent experiments with 6~12 cells per condition per experiment. *p<0.05, **p<0.01, ***p<0.001, by non-parametric student's t-test. MATLAB codes were used for detecting, tracking, and quantifying AF546-Fab' clusters [*Source code 1; 3; 8–14*].

The online version of this article includes the following video and figure supplement(s) for figure 6:

**Figure supplement 1.** Fab'-PLB, but not transferrin-coated PLB (Tf-PLB), induces B-cell receptor (BCR) clustering and phosphorylation.

**Figure supplement 2.** Tracking and analyzing AF546-Fab' clusters in the B-cell contact zone.

**Figure 6—video 1.** Inhibition of B-cell contraction reduces the molecular density within B-cell receptor (BCR) clusters.

https://elifesciences.org/articles/87833/figures#fig6video1

---

top panels, and *Figure 8B*, brown line and symbols). The average MFI ratios of Syk to Fab' in individual clusters increased at a low peak FI range (<140) and did not significantly decrease until Fab' peak FI reached a relatively high range (>280) (*Figure 8B*, black line and symbol). B-cells from cNKO mice exhibited a reduction in the Fab' peak FI of clusters (*Figure 8A*, bottom panels, and *Figure 8C and D*, brown line and symbol) but an increase in the average Syk to Fab' MFI ratios in clusters in a wide range of Fab' peak FI, when compared to flox control B-cells (*Figure 8C and D*, black lines and symbols). However, the average Syk to Fab' MFI ratios of cNKO B-cells decreased in clusters with Fab' peak FI ≥220 and reduced to levels similar to those in flox control B-cells in clusters with Fab' peak FI ≥240 (*Figure 8C and D*, black lines and symbols). These data suggest that increases in the molecular density of BCR clusters, reflected by Fab' peak FI, induce the disassociation of Syk from BCR clusters in both flox control and cNKO B-cells. Our results also show that BCR clusters in cNKO B-cells have significantly higher levels of Syk association than in flox control B-cells, even though they have similar Fab' peak intensities.

We next analyzed the relationship of the pSyk level in individual clusters with BCR molecular density using the method described above. We found that the MFI ratios of pSyk relative to AF546-Fab' gradually decreased with increases in Fab' peak FI in both flox control and cNKO B-cells (*Figure 8E–G*, black lines, and symbols), even though we did not observe an increase in pSyk to Fab' MFI ratio at the low Fab' peak FI range. Similar to the Syk to Fab' MFI ratio, the average pSyk to Fab' MFI ratios of individual clusters in cNKO B-cells were much higher than those in flox control B-cells, except for those at the high Fab' peak FI range (*Figure 8E–H*). To confirm this result, we analyzed equal numbers of pSyk clusters in the same cells, based on the criteria of ≥1.3 fold increase in the peak FI compared to the background outside the contact zone with a diameter of ≥250 nm. Similar to clusters identified by AF546-Fab', the average MFI ratios of pSyk to Fab' in these pSyk clusters decreased with an increase in their Fab' peak FI in both flox control and cNKO B-cells (*Figure 8I–K*, black lines and symbols). Again, the average pSyk to Fab' MFI ratios of pSyk clusters were much higher in cNKO than flox control B-cells, but were reduced to similar levels in clusters with relatively high Fab' peak FI (*Figure 8K*, black line and symbols). Similar reductions of total Syk and pSyk with increasing molecular density of BCR clusters suggest that the disassociation of Syk from BCR clusters, caused by cell contraction-induced increases in molecular density, contributes to BCR signaling attenuation.

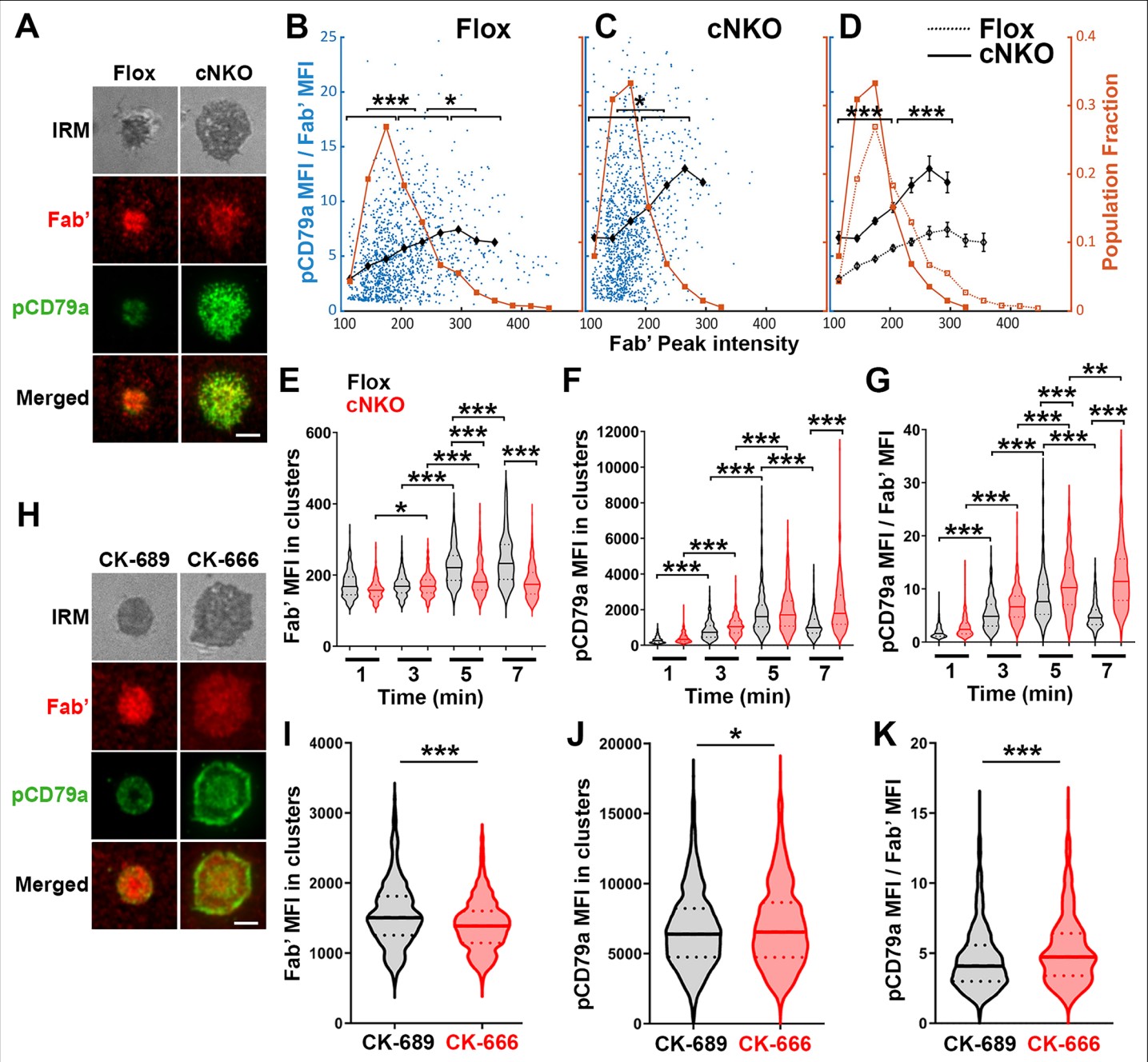

**Figure 7.** Increased molecular density in B-cell receptor (BCR) clusters leads to reductions in BCR phosphorylation. (**A–G**) Flox control and B-cell-specific *N-wasp* knockout mouse (cNKO) B-cells incubated with AF546-Fab'-PLB were fixed at 1, 3, 5, and 7 min, permeabilized, stained for phosphorylated CD79a (pCD79a, Tyr182), and imaged using total internal reflection fluorescence microscopy (TIRF) and interference reflection microscopy (IRM). (**A**) Representative IRM and TIRF images of a flox control versus a cNKO B-cell at 7 min. Scale bars, 2 µm. (**B–D**) Ratios of pCD79a MFI relative to AF546-Fab' MFI were plotted against AF546-Fab' peak FI in individual AF546-Fab' clusters in the contact zone of flox control (**B**), cNKO B-cells (**C**), or flox control and cNKO B-cells overlay (**D**). AF546-Fab' clusters were identified as described in *Figure 6* and *Figure 6—figure supplement 2*. Blue dots represent individual AF546-Fab' clusters with an equal number of clusters from the 4 time points. The black line and diamond symbols represent the average ratios of pCD79a MFI to Fab' MFI in individual BCR-Fab' clusters within the indicated Fab' peak FI range. The brown line and square symbols represent the fraction of the AF546-Fab' clusters out of the total, within the indicated Fab' peak FI range. Clusters were divided into three populations based on their peak AF546-Fab' FI, relatively low (<190), medium (190-280), and high (>280, detected only in contracted cells), and the pCD79a to AF546-Fab' MFI ratios of the three populations were compared (**B** and **C**). Data were generated from three independent experiments with ~20 cells and ≥125 clusters per condition per experiment. *p<0.05, ***p<0.001, by non-parametric student's *t*-test. The *p*-values in (**D**) were corrected using the Benjamini-Hochberg/Yekutieli method for false discovery rate control. (**E–G**) The mean fluorescence intensity (MFI) (± SEM) of AF546-Fab' (**E**) and

*Figure 7 continued on next page*

*Figure 7 continued*

pCD79a (**F**) and the MFI ratio (± SEM) of pCD79a relative to AF546-Fab' (**G**) in individual AF546-Fab' clusters at indicated times were compared between flox control and cNKO B-cells and between different times. (**H–K**) WT B-cells treated with CK-689 or CK-666 after 2 min-incubation with AF546-Fab'-PLB. (**H**) Representative IRM and TIRF images of a CK-689- versus a CK-666-treated B-cell at 7 min. Scale bars, 2 µm. (**I–K**) The MFI (± SEM) of AF546-Fab' (**I**) and pCD79a (**J**) and the MFI ratio (± SEM) of pCD79a relative to AF546-Fab' (**K**) in individual AF546-Fab' clusters were compared between CK-689- and CK-666-treated B-cells after 7 min stimulation. Data points represent individual clusters. Horizontal solid lines in the violin plots represent the mean, while the dotted lines represent the quartiles of the distribution. Data were generated from three independent experiments with ~20 cells per condition per experiment. *p<0.05, ***p<0.001, by non-parametric student's *t*-test. MATLAB codes were used for detecting and quantifying AF546-Fab' clusters [*Source code 1; 10 and 11*].

## Increased BCR molecular density by B-cell contraction promotes disassociation of the inhibitory phosphatase SHIP-1 from BCR microclusters

The inhibitory phosphatase SHIP-1 is essential for B-cell signaling attenuation (*Brauweiler et al., 2000*; *Liu et al., 2011*), suggesting that increases in the molecular density of BCR clusters by B-cell contraction may promote SHIP-1 recruitment. We used the methods described above to address this hypothesis, staining cells for total SHIP-1 (*Figure 9A*) and phosphorylated SHIP-1 (pSHIP-1 Y1020) (*Figure 9E*). We found that the average SHIP-1 to Fab' MFI ratios in both flox control and cNKO B-cells decreased with Fab' peak FI at similar rates (*Figure 9A–D, black lines, and symbols*), even though inhibition of contraction by cNKO reduced the Fab' peak FI of BCR clusters (*Figure 9A–D, brown lines, and symbols*). Notably, the reduction in the SHIP-1 to Fab' MFI ratios with increasing Fab' peak FI occurred at the lowest detectable Fab' peak FI (*Figure 9B–D, brown lines, and symbols*), when the Syk to Fab' MFI ratios increased and were sustained (*Figure 8B–D*). Furthermore, this reciprocal relationship between the SHIP-1 to Fab' MFI ratio and Fab' peak FI continued over the entire Fab' peak FI range. It also remained the same in both flox control and cNKO B-cells (*Figure 9B–D*). These results suggest that SHIP-1 disassociates from BCR clusters as their molecular density increases, and that the SHIP-1 disassociation is more sensitive to the molecular density of BCR clusters than Syk disassociation.

Similar to the relationship of total SHIP-1 with BCR molecular density, the average MFI ratios of pSHIP-1 relative to AF546-Fab' in individual Fab' clusters decreased with increases in Fab' peak FI at similar rates in flox control and cNKO B-cells (*Figure 9E–H, black lines, and symbols*), even though inhibition of contraction by cNKO reduced Fab' peak FI of BCR clusters (*Figure 9E–H, brown lines, and symbols*). The average pSHIP-1 to Fab' MFI ratios in individual pSHIP-1 clusters, detected and analyzed in the same way as pSyk clusters, showed the same decrease with increases in their Fab' peak FI in both flox control and cNKO B-cells (*Figure 9I–K, black lines, and symbols*). Notably, the average pSHIP-1 to Fab' MFI ratios in individual Fab' or pSHIP-1 clusters in flox control and cNKO B-cells were at similar levels at the same Fab' peak FI ranges (*Figure 9H and K, black lines, and symbols*). These results indicate that contraction-induced molecular density increases within individual BCR clusters do not induce preferential recruitment of SHIP-1; instead, it promotes the disassociation of SHIP-1 from BCR clusters.

## Discussion

When binding membrane-associated antigens, naive follicular B-cells undergo actin-mediated spreading followed by a contraction, which amplifies BCR signaling and promotes immunological synapse formation (*Fleire et al., 2006*; *Harwood and Batista, 2011*). We previously showed that B-cell contraction after spreading on antigen-presenting surfaces promotes BCR signaling attenuation (*Liu et al., 2013a*; *Seeley-Fallen et al., 2022*). However, how the actin cytoskeleton reorganizes as B-cells transition from spreading to contraction and how B-cell contraction downregulates BCR signaling have been elusive. Here, we demonstrate that inner F-actin foci formed at the contact zone distal to the lamellipodial F-actin network promote B-cells to switch from spreading to contraction. These inner foci are derived from the lamellipodial F-actin network that mediates spreading, are generated by N-WASP- but not WASP-activated Arp2/3-mediated branched actin polymerization, and facilitate NMII recruitment. N-WASP-activated actin polymerization coordinates with NMII to form actomyosin ring-like structures, enabling B-cell contraction. B-cell contraction increases BCR molecular density in

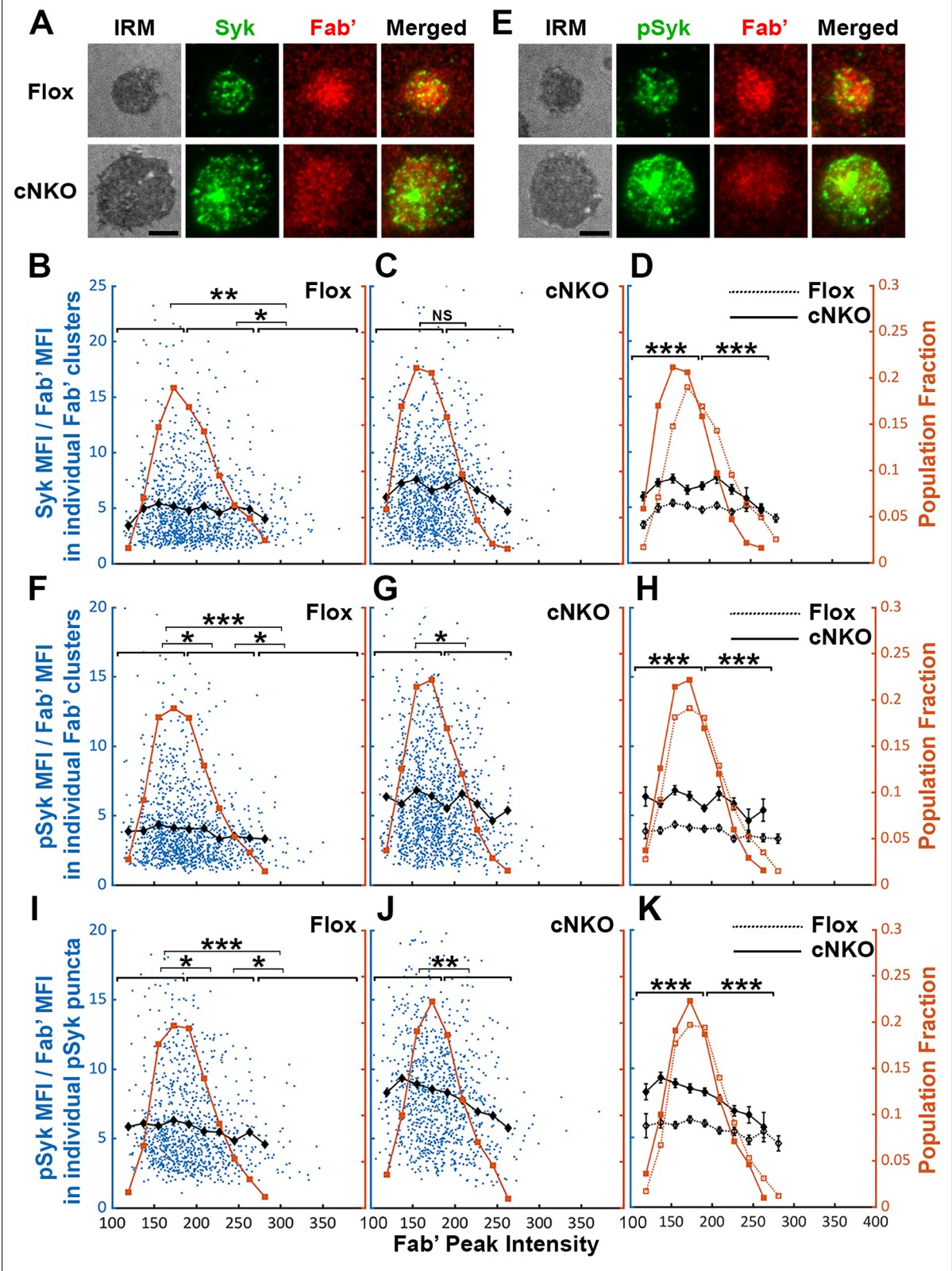

**Figure 8.** Effects of BCR-Fab' density on the association of spleen tyrosine kinase (Syk) with BCR-Fab' clusters and its phosphorylation. Primary B-cells from flox control and B-cell-specific *N-wasp* knockout mice (cNKO) mice were incubated with AF546-Fab'-PLB at 37 °C, fixed at 3 or 7 min, stained for total Syk and phosphorylated Syk (pSyk Y519/520), and imaged by interference reflection microscopy (IRM) and total internal reflection fluorescence microscopy (TIRF). (**A**) Representative IRM and TIRF images of Syk staining in a flox control versus a cNKO B-cell at 7 min (Scale bars, 2 µm). (**B–D**) Ratios

*Figure 8 continued on next page*

*Figure 8 continued*

of Syk MFI relative to AF546-Fab' MFI were plotted against AF546-Fab' peak fluorescence intensity (FI) in individual AF546-Fab' clusters in the contact zone of Flox control (**B**) and cNKO B-cells (**C**) and their overlay (**D**). AF546-Fab' clusters were identified as described in *Figure 6* and *Figure 6—figure supplement 2* from an equal number of cells after 3- and 7 min stimulation. Blue dots represent individual AF546-Fab' clusters with an equal number of clusters from each time point. The black line and diamond symbols represent the average ratios of Syk MFI to Fab' MFI in individual AF546-Fab' clusters at indicated Fab' peak FI ranges. The brown line and square symbols represent the fraction of the AF546-Fab' clusters out of the total at indicated Fab' peak FI ranges. (**E**) Representative IRM and TIRF images of pSyk staining in a flox control versus a cNKO B-cell at 7 min (Scale bars, 2 μm). (**F–H**) Ratios of pSyk MFI relative to AF546-Fab' MFI were plotted against AF546-Fab' peak FI in individual AF546-Fab' clusters in the contact zone of flox control (**F**) and cNKO B-cells (**G**) and their overlay (**H**). Blue dots represent individual AF546-Fab' clusters with an equal number of clusters from each time point. The black line and diamond symbols represent the average ratios of pSyk MFI to Fab' MFI in individual AF546-Fab' clusters at indicated Fab' peak FI ranges. The brown line and square symbols represent the fraction of the AF546-Fab' clusters out of the total at indicated Fab' peak FI ranges. (**I–K**) MFI ratios of pSyk relative to AF546-Fab' were plotted against AF546-Fab' peak FI in individual pSyk puncta in the contact zone of Flox control (**I**) and cNKO B-cells (**J**) and their overlay (**K**). pSyk puncta were identified using the criteria: FI ≥1.3 fold of the background outside the B-cell contact zone and diameter ≥250 nm. Blue dots represent individual pSyk puncta with an equal number of clusters from each time point. The black line and diamond symbols represent the average ratios of pSyk MFI to Fab' MFI in individual pSyk puncta at indicated Fab' peak FI ranges. The brown line and square symbols represent the fraction of the pSyk puncta out of the total at indicated Fab' peak FI ranges. Clusters were divided into three populations based on their peak AF546-Fab' FI, relatively low (<190), medium (190-280), and high (>280, detected only in contracted cells), and the Syk (**B** and **C**) or pSyk (**F**, **G**, **I**, and **J**) to AF546-Fab' MFI ratios of the three populations were compared. Data were generated from three independent experiments with ~23 cells and ≥125 clusters per condition per experiment. *p<0.05, **p<0.01, ***p<0.001, by non-parametric student's *t*-test, between AF546-Fab' cluster group with different Fab' peak FI ranges. The p-values in (**D**, **H**, and **K**) were corrected using the Benjamini-Hochberg/Yekutieli method for false discovery rate control. MATLAB codes were used for detecting and quantifying AF546-Fab' clusters [*Source code 1; 10 and 11*].

existing clusters, which promotes the disassociation of both the stimulatory kinase Syk and the inhibitory phosphatase SHIP-1, leading to signaling attenuation.

One significant finding of this study is that Arp2/3-mediated polymerization of branched actin is required for B-cell contraction. Arp2/3-mediated branched actin polymerization is known to drive B-cell spreading and to create actin centripetal flow at the contact zone between B-cells and antigen-presenting surface (*Bolger-Munro et al., 2019*). The actin structure that supports B- and T-cell spreading to form the immunological synapse with antigen-presenting cells is similar to lamellipodial F-actin networks found in adherent cells (*Bunnell et al., 2001*; *Koestler et al., 2008*; *Bolger-Munro et al., 2019*). Lamellipodial F-actin networks consist primarily of branched actin filaments polymerizing against the plasma membrane interspersed with bundled actin filaments (*Krause and Gautreau, 2014*; *Skau and Waterman, 2015*). Contractile actin structures, such as stress fibers, are typically generated from bundled actin filaments, as observed in adherent and migrating cells (*Levayer and Lecuit, 2012*; *Tojkander et al., 2015*; *Hammer et al., 2019*). Here, we show that inner F-actin foci, generated by Arp2/3-mediated actin polymerization, transition B-cells from spreading to contraction. Furthermore, these inner F-actin foci are directly derived from lamellipodial F-actin networks. Based on the observed dynamics, we infer that instead of polymerizing against the plasma membrane, Arp2/3 appears to nucleate actin polymerization in the opposite direction, sustaining actin foci and their movement away from the lamellipodia. The natural retraction of lamellipodia may contribute to B-cell contraction, but the requirement of inner actin foci argues against it as a primary mechanism for B-cell contraction. This study does not exclude the involvement of bundled actin filaments. Based on the requirement of formin-activated bundled actin filaments for lamellipodial F-actin networks and B-cell spreading (*Wang et al., 2022*), we can speculate that bundled actin filaments may play a role in the formation and movement of these inner F-actin foci, as well as in NMII recruitment.

Our work provides new insights into distinct functions of the actin nucleation-promoting factors WASP and N-WASP in controlling cell morphology and signaling. Immune cells, including B-cells, express both hematopoietic-specific WASP and the ubiquitous homolog of WASP, N-WASP. These two share high sequence homology, activation mechanism, and Arp2/3 activation function (*Padrick and Rosen, 2010*). We previously identified distinct functions of these two factors unique to B-cells. While both are required for B-cell spreading, N-WASP plays a unique role in B-cell contraction. *Wasp* and *N-wasp* double knockout B-cells fail to spread on antigen-presenting surfaces, but B-cell-specific *N-wasp* knockout enhances B-cell spreading and delays B-cell contraction (*Liu et al., 2013a*). WASP and N-WASP appear to have a competitive relationship in B-cells, suppressing each other's activation (*Liu et al., 2013a*). WASP is activated first during B-cell spreading and is primarily localized at the periphery of the B-cell contact zone. N-WASP is activated later during B-cell contraction and scattered

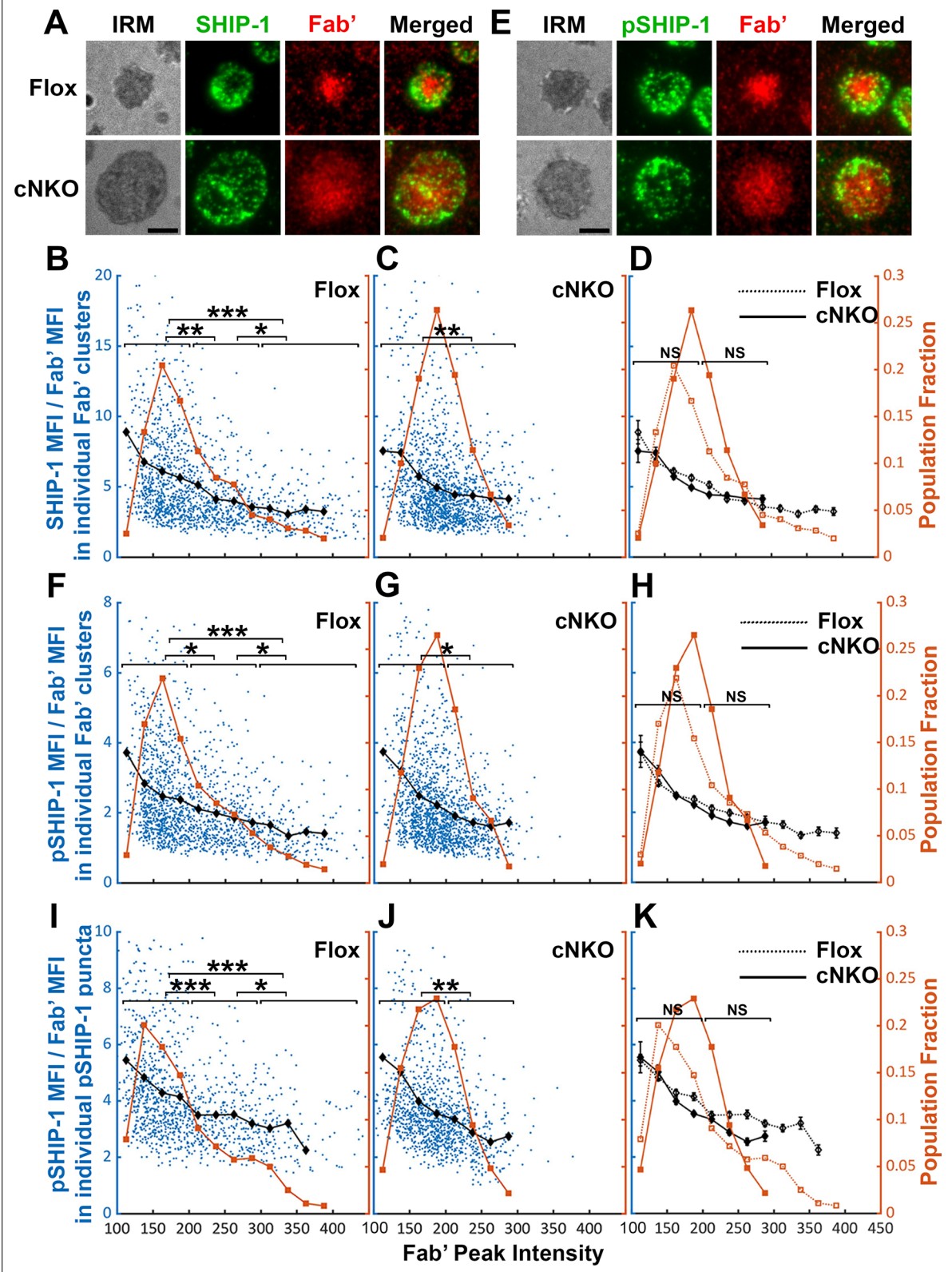

**Figure 9.** The effects of BCR-Fab' density on the association of phosphatidylinositol-5 phosphatase-1 (SHIP-1) with BCR-Fab' clusters and its phosphorylation. Primary B-cells from flox control and B-cell-specific *N-wasp* knockout mice (cNKO) mice were incubated with AF546-Fab'-PLB at 37 °C, fixed at 3 or 7 min, stained for total SHIP-1 and phosphorylated SHIP-1 (pSHIP-1 Tyr1020), and imaged by interference reflection microscopy (IRM) and total internal reflection fluorescence microscopy (TIRF). (**A**) Representative IRM and TIRF images of SHIP-1 staining in a flox control versus a cNKO B-cell

*Figure 9 continued on next page*

*Figure 9 continued*

at 7 min (Scale bars, 2 μm). (**B–D**) Ratios of SHIP-1 MFI relative to AF546-Fab' MFI were plotted against AF546-Fab' peak fluorescence intensity (FI) in individual AF546-Fab' clusters in the contact zone of Flox control (**B**) and cNKO B-cells (**C**). An overlay of flox control and cNKO plots is also shown (**D**). AF546-Fab' clusters were identified as described in *Figure 6* and *Figure 6—figure supplement 2* from an equal number of cells after 3 and 7 min stimulation. Blue dots represent individual AF546-Fab' clusters with an equal number of clusters from each time point. The black line and diamond symbols represent the average ratios of SHIP-1 MFI to Fab' MFI in individual AF546-Fab' clusters at indicated Fab' peak FI ranges. The brown line and square symbols represent the fraction of the AF546-Fab' clusters out of the total at indicated Fab' peak FI ranges. (**E**) Representative IRM and TIRF images of pSHIP-1 staining in a flox control versus a cNKO B-cell at 7 min (Scale bars, 2 μm). (**F–H**) Ratios of pSHIP-1 MFI relative to AF546-Fab' MFI were plotted against AF546-Fab' peak FI in individual AF546-Fab' clusters in the contact zone of flox control (**F**) and cNKO B-cells (**G**) and their overlay (**H**). Blue dots represent individual AF546-Fab' clusters with an equal number of clusters from each time point. The black line and diamond symbols represent the average ratios of pSHIP-1 MFI to Fab' MFI in individual AF546-Fab' clusters at indicated Fab' peak FI ranges. The brown line and square symbols represent the fraction of the AF546-Fab' clusters out of the total at indicated Fab' peak FI ranges. (**I–K**) Mean fluorescence intensity (MFI) ratios of pSHIP-1 relative to AF546-Fab' were plotted against AF546-Fab' peak FI in individual pSHIP-1 puncta in the contact zone of Flox control (**I**) and cNKO B-cells (**J**) and their overlay (**K**). pSHIP-1 puncta were identified using the criteria: FI ≥1.5 fold of the background outside the B-cell contact zone and diameter ≥250 nm. Blue dots represent individual pSHIP-1 puncta with an equal number of clusters from each time point. The black line and diamond symbols represent the average ratios of pSHIP-1 MFI to Fab' MFI in individual pSHIP-1 puncta at indicated Fab' peak FI ranges. The brown line and square symbols represent the fraction of the total pSHIP-1 puncta at indicated Fab' peak FI ranges. Clusters were divided into three populations based on their peak AF546-Fab' FI, relatively low (<200), medium (200-300), and high (>300, detected only in contracted cells), and the SHIP-1 (**B** and **C**) or pSHIP-1 (**F**, **G**, **I**, and **J**) to AF546-Fab' MFI ratios of the three populations were compared. Data were generated from three independent experiments with ~23 cells and ≥125 clusters per condition per experiment. *p<0.05, **p<0.01, ***p<0.001, by non-parametric student's *t*-test, between AF546-Fab' cluster group with different Fab' peak FI ranges. The p-values in (**D**, **H**, and **K**) were corrected using the Benjamini-Hochberg/Yekutieli method for false discovery rate control. MATLAB codes were used for detecting and quantifying AF546-Fab' clusters *Source code 1; 10 and 11*.

across the B-cell contact zone (*Liu et al., 2013a*). Here, we reveal the exact role of N-WASP in B-cell contraction – to generate inner F-actin foci from lamellipodial F-actin networks by activating Arp2/3-mediated actin polymerization. N-WASP- but not WASP-activated actin polymerization prolongs the relative lifetime of F-actin foci and facilitates their inward motion toward the center of the contact zone. The delayed activation time and the location in the interior of the contact zone likely give N-WASP a unique opportunity to generate inner F-actin foci. As cNKO only delays and reduces but does not block B-cell contraction and inner F-actin foci formation, other actin factors may be involved, such as the WASP-family verprolin-homologous protein (WAVE) (*Rotty et al., 2013*).

Our recently published data show that NMII is required for B-cell contraction (*Seeley-Fallen et al., 2022*). Activated NMII is recruited to the B-cell contact zone in a SHIP-1-dependent manner. NMII levels reach a plateau at the beginning of B-cell contraction, and NMII forms a peripheral ring surrounding the contact zone during B-cell contraction. Here, we further show that in addition to SHIP-1, N-WASP but not WASP is also involved in NMII recruitment and NMII ring-like structure formation, probably by initiating the generation of inner F-actin foci. Collectively, these findings suggest that SHIP-1 coordinates with N-WASP to recruit NMII. Our previous finding that SHIP-1 promotes B-cell contraction by facilitating N-WASP activation (*Liu et al., 2011*; *Liu et al., 2013a*) supports this notion. However, it is surprising that the recruitment of NMII activated by BCR signaling to the contact zone is associated with inhibitory signaling molecules. BCR downstream signaling, including $Ca^{2+}$ fluxes and Rho-family GTPase activation, likely activates NMII motor activity (*Vicente-Manzanares et al., 2009*). The activation switches NMII from the incompetent folded conformation to the competent extended conformation, enabling NMII molecules to bind to F-actin and assemble into contractible bipolar filaments and stacks (*Matsumura, 2005*). A denser F-actin organization has been shown to increase NMII filament stacking (*Fenix and Burnette, 2018*). Here, we showed that the inner F-actin foci generated by N-WASP-activated Arp2/3 are more stable and denser and thus likely promote NMII binding and stacking more efficiently than F-actin generated by WASP-activated Arp2/3. Indeed, our kymograph analysis revealed similar time windows and spatial locations for NMII accumulation and for generating inner F-actin foci from lamellipodial F-actin, supporting our hypothesis. While NMII recruitment is facilitated by inner F-actin foci, the motor activity of NMII is required for the formation of the ring-like structures of inner F-actin foci in return, likely by driving their centripetal movement, thereby providing mechanical feedback for actin reorganization. However, the structural organization of inner F-actin foci and recruited NMII and their coordinated dynamics require further analysis with higher-resolution microscopy techniques.

*Wang et al., 2022* recently showed that B-cells generate actomyosin arcs when interacting with membrane-associated antigen in the presence of adhesion molecules expressed on professional antigen-presenting cells. In contrast to T-cells, B-cells can also respond to membrane-associated antigens without the help of adhesion molecules, when engaging antigens on pathogenic cells, like bacteria, parasites, and cancer cells. While the relationship between actomyosin arcs and inner F-actin foci remains to be explored, the coordinated formation of inner F-actin foci and NMII ring-like structures revealed in our study suggests that these structures are likely the precursors of actomyosin arcs. These structures can potentially be enhanced and matured by adhesion molecule interactions between B-cells and antigen-presenting membranes.

Our findings suggest an increase in the molecular density within BCR clusters or B-cell synapses as one of the mechanisms by which B-cell contraction promotes BCR signaling attenuation. During contraction, the molecular density of individual BCR clusters, measured by their MFI and peak FI of BCR clustered Fab', increased faster than during the spreading and post-contraction phases. Surprisingly, increases in molecular density induced disassociation of both the stimulatory kinase Syk and the inhibitory phosphatase SHIP-1, contradicting the existing dogma of sequential association of stimulatory and inhibitory signaling molecules with the BCR (*Franks and Cambier, 2018*). Interestingly, the disassociation of SHIP-1 is much more sensitive to BCR molecular density than Syk disassociation. Syk does not disassociate until the molecular density of BCR clusters reaches the top 10% range, while SHIP-1 disassociates from BCR clusters with increases over the entire detectable range of molecular densities. When B-cell contraction is inhibited, the molecular density of BCR clusters is reduced to a range with limited Syk disassociation and normal SHIP-1 disassociation, probably resulting in higher Syk to SHIP-1 molecular ratios in BCR clusters than contracting B-cells, increasing BCR signaling levels. The consistency of the data from AF546-Fab' and pSyk or pSHIP-1 clusters supports this notion. However, the resolution of our widefield TIRF imaging limited our ability to examine nascent BCR clusters that actively recruit signaling molecules. Available antibodies did not allow us to simultaneously stain Syk and SHIP-1 or their phosphorylated forms and determine their relative levels in the same cluster. The timing and dynamics of Syk and SHIP-1 recruitment to individual BCR clusters and the direct relationship between Syk and SHIP-1 association and disassociation with BCR clusters and their phosphorylation remain to be further investigated.

How the molecular density within BCR clusters promotes disassociation of signaling molecules is unknown. We speculate that molecular crowding may displace signaling molecules out of BCR clusters. Supporting this possibility, disassociation of the 145 kDa SHIP-1 is likely to be more sensitive to changes in the molecular density of BCR clusters, compared to disassociation of the 72 kDa Syk. However, these hypotheses need to be further examined. An interesting finding of this study is that the pCD79a and pSyk levels were higher in individual BCR clusters of cNKO B-cells than those of flox control B-cells, even when BCR clusters had similar molecular densities. These data suggest additional mechanisms for B-cell contraction to promote BCR signaling attenuation. In addition to increasing molecular density within BCR clusters, the contractile forces generated by actomyosin rings on the B-cell membrane and BCR clusters may cause changes to the conformations of both BCRs and associated signaling molecules and their lateral interactions in the membrane. These changes may favor the disassociation rather than the association of signaling molecules with BCR clusters. Recently solved molecular assembly structures of both human and mouse BCRs (*Dong et al., 2022*; *Su et al., 2022*) support conformational and lateral molecular interaction changes as possible mechanisms for BCR signaling regulation. Similar to the differential impact of the molecular density on the disassociation of Syk and SHIP-1 from BCR clusters, B-cell contraction may differentially affect the interaction of surface BCRs with cytoplasmic and membrane-anchored signaling molecules, such as the lipidated and lipid raft-resident Src kinase Lyn that is responsible for phosphorylating both the immunoreceptor tyrosine-based activation motif (ITAM) of CD79a and the immunoreceptor tyrosine-based inhibitory motif (ITIM) that SHIP-1 binds to (*Franks and Cambier, 2018*).

Following antigen-induced BCR signaling at immunological synapses, B-cells internalize BCR-captured antigen for processing and presentation. Actomyosin is required for BCR-mediated endocytosis of antigens, particularly surface- and membrane-associated antigens (*Natkanski et al., 2013*; *Hoogeboom et al., 2018*; *Maeda et al., 2021*). NMII-mediated traction forces pull BCR-bound antigen off presenting surfaces for endocytosis in an affinity-dependent manner. We have previously shown that when antigens are tightly attached to a surface, high-affinity binding of the BCR to antigens tears

the B-cell membrane in an NMII-dependent manner, which triggers lysosome exocytosis and lysosomal enzyme-mediated cleavages of antigen from the associated surface, allowing antigen endocytosis (*Maeda et al., 2021*). While actomyosin plays essential roles in both B-cell contraction and the subsequent antigen endocytosis, whether the actomyosin structure responsible for B-cell contraction also mediates BCR endocytosis remains an interesting and open question. Our early finding that N-WASP is required for BCR endocytosis (*Liu et al., 2013a*) supports this notion. BCR endocytosis can reduce B-cell surface signaling by removing the BCR from the cell surface, transitioning the BCR into an intracellular signaling state (*Chaturvedi et al., 2008*).

The results presented here have revealed novel insights into the mechanisms underlying actin-facilitated signaling attenuation of the BCR. Taking the previous and current data of our and other labs together, we propose a new working model for such actin-mediated signaling downregulation (*Figure 10*). Upon encountering membrane-associated antigens, such as antigen presented by follicular dendritic cells and on the surfaces of pathogens, mature follicular B-cells undergo rapid spreading, primarily driven by WASP-mediated branched actin polymerization, which maximizes B-cell contact with antigen-presenting surfaces and BCR-antigen engagement, amplifying signaling. Following maximal spreading, N-WASP distal to lamellipodial networks activates Arp2/3-mediated branched actin polymerization, which initiates the formation of inner F-actin foci from lamellipodia towards the center of the B-cell contact zone (*Figure 10*). NMII is then preferentially recruited to these relatively stable inner foci, which promotes the centripetal movement of inner F-actin foci and the maturation of ring-like actomyosin structures, enabling B-cell contraction. B-cell contraction pushes the BCR microclusters formed during B-cell spreading to the center of the contact zone, increasing their molecular density. Increased molecular density promotes the disassociation of signaling molecules from BCR clusters, probably due to crowding and conformational changes, leading to signal downregulation (*Figure 10*). Inhibitory signaling molecules likely activate the actin reorganization that drives B-cell contraction by promoting the activation and recruitment of NMII to the B-cell contact zone (*Liu et al., 2013a*; *Seeley-Fallen et al., 2022*) by hitherto unknown mechanisms. The actomyosin structures responsible for B-cell contraction may further drive B-cells to internalize engaged antigen, further downregulating signaling at the B-cell surface and initiating antigen processing and presentation. Thus, actin reorganization downstream of inhibitory phosphatases reinforces signaling attenuation by driving B-cell contraction.

# Materials and methods

**Key resources table**

| Reagent type (species) or resource | Designation | Source or reference | Identifiers | Additional information |
|---|---|---|---|---|
| Strain (*Mus musculus*) | C57BL/6 (WT) | Jackson Laboratories | JAX stock #000664 | Primary cells |
| Strain (*Mus musculus*) | *Wasp*$^{-/-}$ mice (WKO) | Jackson Laboratories | JAX stock #019458 | Primary cells |
| Strain (*Mus musculus*) | *Cd19*$^{Cre/+}$*N-wasp*$^{flox/flox}$ mice (cNKO) | Lisa Westerberg laboratory | | Primary cells |
| Strain (*Mus musculus*) | *N-wasp*$^{flox/flox}$ (Flox control) | Lisa Westerberg laboratory | | Primary cells |
| Strain (*Mus musculus*) | LifeAct-GFP | Roberto Weigert laboratory | | Primary cells |
| Strain (*Mus musculus*) | LifeAct-RFP | Klaus Ley laboratory | | Primary cells |
| Strain (*Mus musculus*) | GFP-NMIIA | Robert Adelstein laboratory | | Primary cells |
| Chemical compound | 1,2-dioleoyl-sn-glycero-3-phosphocholine | Avanti Polar Lipids | 850375 P | Liposomes (5 mM) |
| Chemical compound | 1,2-dioleoyl-sn-glycero-3-phosphoethanolamine-cap-biotin | Avanti Polar Lipids | 870273 C | Liposomes (50 µM) |

*Continued on next page*

*Continued*

| Reagent type (species) or resource | Designation | Source or reference | Identifiers | Additional information |
|---|---|---|---|---|
| Other | Streptavidin | Jackson Immuno Research | 016-000-084 | (1 µg/ml) |
| Antibody | F(ab')$_2$ fragment of goat IgG anti-mouse Ig(G+M) (goat polyclonal) | Jackson Immuno Research | 115-006-068 | - |
| Antibody | Cy3-Fab fragment of goat anti-mouse IgG + M (goat polyclonal) | Jackson Immuno Research | 115-167-020 | (2.5 µg per 1x10$^6$ cells) |
| Chemical compound | 2-Mercaptoethylamine HCL | Thermo Fisher Scientific | 20408 | (50 mM) |
| Other | EZ-Link Maleimide-PEG$_2$-biotin | Thermo Fisher Scientific | A39261 | (20 mM per mM of protein) |
| Commercial kit | Alexa Fluor 546 antibody labeling kit | Thermo Fisher Scientific | A20183 | - |
| Other | Biotinylated holo-transferrin (Tf) | Jackson Immuno Research | 015-060-050 | - |
| Antibody | Rat IgG$_{2b}$ anti-mouse CD90.2 (Thy1.2) (rat monoclonal) | Biolegend | 105351 | (1 µl per 2x10$^6$ cells) |
| Chemical compound | CK-689 | Millipore Sigma | 182517–25 MG | Inhibitor control (50 µM) |
| Chemical compound | CK-666 | Millipore Sigma | SML0006-5MG | Arp2/3 inhibitor (50 µM) |
| Chemical compound | Wiskostatin | Millipore Sigma | W2270-5MG | N-WASP inhibitor (10 µM) |
| Chemical compound | Blebbistatin | Cayman Chemicals | 13013 | NMII inhibitor (50 µM) |
| Antibody | Rabbit IgG anti-mouse Arp2 antibody (rabbit polyclonal) | Abcam | ab47654 | IF (1:100) |
| Antibody | Rabbit IgG anti-mouse pWASP antibody (rabbit polyclonal) | Thermo Fisher Scientific | PA5-105572 | IF (1:100) |
| Antibody | Rabbit IgG anti-mouse pN-WASP antibody (rabbit polyclonal) | Thermo Fisher Scientific | PA5-105307 | IF (1:100) |
| Antibody | AF488-goat IgG anti-rabbit IgG antibody (goat polyclonal) | Thermo Fisher Scientific | A-11034 | IF (1:200) |
| Antibody | AF546-goat IgG anti-rabbit IgG antibody (goat polyclonal) | Thermo Fisher Scientific | A-11035 | IF (1:200) |
| Other | Acti-stain-488 Phalloidin | Cytoskeleton | PHDG1-A | IF (200 nM) |
| Other | Acti-stain-555 Phalloidin | Cytoskeleton | PHDH1-A | IF (200 nM) |
| Antibody | Rabbit IgG anti-mouse NMIIA antibody (rabbit polyclonal) | Abcam | ab75590 | IF (1:100) |
| Antibody | Rabbit IgG anti-mouse pCD79a (Y182) (rabbit monoclonal) | Cell Signaling Technology | 14732 S | IF (1:100) |
| Antibody | Rabbit IgG anti-mouse pSyk (Y519/520) (rabbit monoclonal) | Cell Signaling Technology | 2710 S | IF (1:100) |
| Antibody | Rabbit IgG anti-mouse pSHIP-1 (Y1020) (rabbit polyclonal) | Cell Signaling Technology | 3941 S | IF (1:100) |
| Antibody | Rabbit IgG anti-mouse Syk (rabbit polyclonal) | Thermo Fisher Scientific | PA5-17812 | IF (1:100) |
| Antibody | Rabbit IgG anti-mouse SHIP-1 (rabbit polyclonal) | Thermo Fisher Scientific | PA5-115894 | IF (1:100) |
| Software, algorithm | MATLAB | MathWorks | R2022a | |
| Software, algorithm | Fiji-ImageJ | Fiji organization | Version 2.9.0/1.53t | |
| Software, algorithm | GraphPad Prism | GraphPad | Version 9.2.0 | |

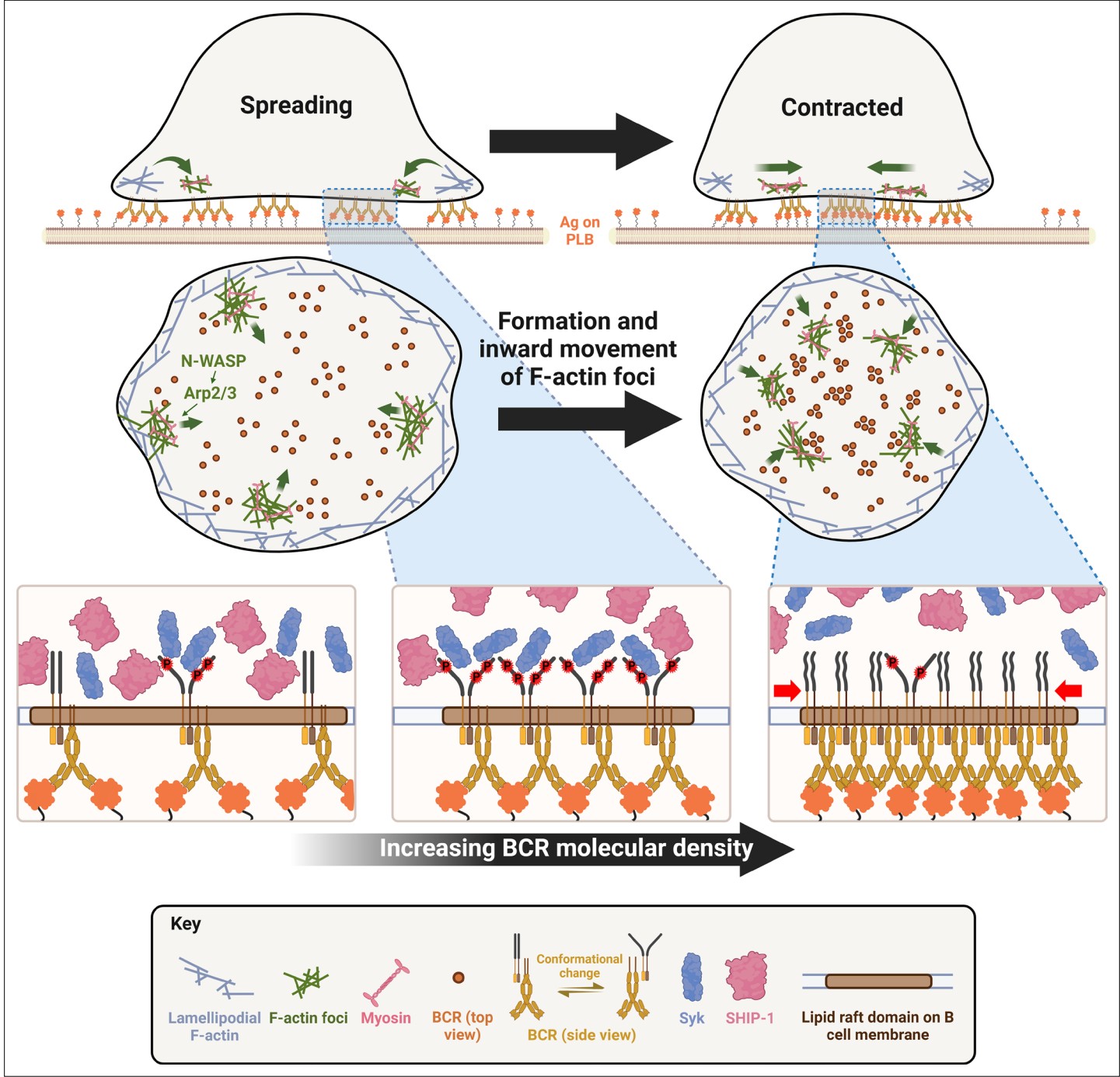

**Figure 10.** A working model for cell contraction-mediated B-cell receptor (BCR) signaling attenuation. When encountering membrane-associated antigen, mature follicular B-cells undergo rapid spreading, primarily driven by WASP-mediated branched actin polymerization, which maximizes B-cell contact with antigen-presenting surface and BCR-antigen engagement, amplifying signaling. Upon reaching maximal spreading, neuronal-WASP (N-WASP) distal to lamellipodial networks activates Arp2/3-mediated branched actin polymerization, which initiates the generation of inner F-actin foci from lamellipodia towards the center of the B-cell contact zone. Non-muscle myosin II (NMII) is then preferentially recruited to these relatively stable inner foci, which in turn promotes the centripetal movement of inner F-actin foci and the maturation of ring-like actomyosin structures, enabling B-cell contraction. B-cell contraction pushes the BCR microclusters formed during B-cell spreading to the center of the contact zone, increasing their molecular density. Increased molecular density promotes the disassociation of signaling molecules from BCR clusters, probably due to crowding and conformational changes, leading to signaling attenuation.

## Mice and B-cell isolation

Wild-type (WT) C57BL/6 mice and *Wasp* knockout (WKO) mice on the C57BL/6 background were purchased from Jackson Laboratories. C57BL/6 mice expressing LifeAct-GFP transgene (*Riedl et al., 2010*) were kindly provided by Dr. Roberto Weigert at the National Cancer Institute, USA. C57BL/6 mice expressing the LifeAct-RFP transgene were kindly provided by Dr. Klaus Ley at the La Jolla Institute for Allergy & Immunology. C57BL/6 mice expressing the GFP-non-muscle myosin IIA (NMIIA) transgene were kindly provided by Dr. Robert Adelstein at the National Heart, Lung, and Blood Institute, USA. WKO mice were bred with LifeAct-GFP mice to obtain LifeAct-GFP-expressing WKO mice. GFP-NMIIA mice were crossed with LifeAct-RFP mice to generate mice expressing both transgenes. *N-wasp^{flox/flox}* on a 129Sv background (*Cotta-de-Almeida et al., 2007*) were kindly provided by Dr. Lisa Westerberg at Karolinska Institute, Sweden. B-cell-specific *N-wasp* knockout mice (cNKO, *Cd19^{Cre/+}N-wasp^{flox/flox}*) and floxed littermate controls (*N-wasp^{flox/flox}*) were obtained by breeding *N-wasp^{flox/flox}* mice with *Cd19^{Cre/+}* mice on a C57BL/6 background.

Primary B-cells were isolated from the spleens of 6- to 18-week-old male or female mice, using a previously published protocol (*Sharma et al., 2009*). Briefly, mononuclear cells were isolated by Ficoll density-gradient centrifugation (Millipore Sigma), and T-cells were eliminated by complement-mediated cytolysis with anti-mouse CD90.2 mAb (BD Biosciences) and guinea pig complement (Innovative Research Inc). Monocytes and dendritic cells were eliminated by panning at 37 °C and 5% $CO_2$. Isolated B-cells were kept on ice in DMEM (Lonza) supplemented with 0.6% BSA (Thermo Fisher Scientific). All work involving mice was approved by the Institutional Animal Care and Usage Committee of the University of Maryland and followed the public health service policy on humane care and use of laboratory animals.

## Pseudo-antigen-coated planar lipid bilayers

Planar lipid bilayers (PLB) were prepared using a previously described method (*Dustin et al., 2007*; *Liu et al., 2012a*). Briefly, liposomes were generated from a mixture of 5 mM (total concentration) 1,2-dioleoyl-sn-glycero-3-phosphocholine and 1,2-dioleoyl-sn-glycero-3-phosphoethanolamine-cap-biotin (Avanti Polar Lipids) at a 100:1 molar ratio by sonication. Glass coverslips, cleaned overnight with Piranha solution (KMG chemicals), were attached to eight-well chambers (Lab-Tek) and incubated with liposomes (0.05 mM) in PBS for 20 min at room temperature and washed with PBS. The chambers were then incubated with 1 µg/ml streptavidin (Jackson ImmunoResearch Laboratories) for 10 min, washed with PBS, and then incubated with 10 µg/ml mono-biotinylated Fab' fragment of goat anti-mouse Ig(G+M) (pseudo-antigen) (Fab'-PLB) for 10 min, followed by PBS wash. For transferrin-coated PLB (Tf-PLB), 16 µg/ml biotinylated holoTF was used. To visualize Fab' clustering, a mixture of 0.5 µg/ml Alexa Fluor (AF) 546-labeled and 9.5 µg/ml unlabeled mono-biotinylated Fab' fragment of goat anti-mouse Ig(G+M) was used. The lateral mobility of AF546-Fab' on the PLB was tested using fluorescence recovery after photobleaching (FRAP) to ensure ≥85% AF546 fluorescence recovery within 1 min after photobleaching using a Zeiss LSM 710 equipped with a 60 X oil-immersion objective.

Mono-biotinylated Fab' fragments were generated as previously described (*Liu et al., 2011*). Briefly, the disulfide bond linking the two Fab fragments of F(ab')$_2$ goat anti-mouse Ig(G+M) was reduced using 2-mercaptoethylamine HCL (Thermo Fisher Scientific) and biotinylated by using maleimide-PEG$_2$-biotin (Thermo Fisher Scientific). Mono-biotinylated Fab' fragments of goat anti-mouse Ig(G+M) were labeled with AF546 using an Alexa Fluor 546 antibody labeling kit (Thermo Fisher Scientific). The molar ratio of AF546 to Fab' in the AF546-labeled Fab' was ~2 determined by a Nanodrop spectrophotometer (Nanodrop Technologies).

## Total internal fluorescence microscopy

To visualize molecules proximal to interacting sites between B-cells and Fab'- or Tf-PLB, we utilized total internal reflection fluorescence microscopy (TIRF) and interference reflection microscopy (IRM). Images were acquired using a Nikon TIRF system on an inverted microscope (Nikon TE2000-PFS, Nikon Instruments Inc) equipped with a 60 X, NA 1.49 Apochromat TIRF objective (Nikon), a Coolsnap HQ2 CCD camera (Roper Scientific), and two solid-state lasers of wavelength 491 and 561 nm. IRM, AF488, and AF546 images were acquired sequentially.

The plasma membrane area of B-cells contacting PLB was determined using IRM images and custom MATLAB codes [*Source code 1–3 and 5*]. Whether a B-cell contracted or not was determined

using the area versus time plots, wherein if a B-cell's contact zone area reduced by ≥5% for at least 10 s after reaching a maximum value, it was classified as contracting. To image intracellular molecules, B-cells were incubated with Fab'-PLBs for varying lengths of time in PBS at 37 °C, fixed with 4% paraformaldehyde, permeabilized with 0.05% saponin, and stained for various molecules. For live-cell imaging, B-cells were pre-warmed to 37 °C and imaged as soon as B-cells were dropped into cover-slip chambers coated with Fab'-PLB containing PBS in a humidity chamber at 37 °C, at 2 s per frame, up to 7 min. All images from multiple independent experiments were analyzed using Fiji ImageJ and custom MATLAB scripts [*Source code 4–8; 10–14* ]. Acquired FI data were normalized to the one with the lowest FI.

### Inhibitors

CK-666 (50 µM, Millipore Sigma) was used to perturb Arp2/3 activity (*Nolen et al., 2009*), and its non-functional derivative CK-689 (50 µM, Millipore Sigma) as a control. CK-666 or CK-689 was added at either 0 min, the start of incubation with Fab'-PLB, or at 2 min when most B-cells reached maximum spreading under the described condition. Notably, B-cells take approximately 1–1.5 min to land on PLB. The time when B-cells were added to Fab'-PLB is referred to as 0 min, and the time when B-cells landed on PLB as the start of spreading. The effectiveness of CK-666 was determined by its inhibitory effects on the recruitment of Arp2/3, stained by an anti-Arp2 antibody (Abcam), to the B-cell contact zone using TIRF (*Figure 1—figure supplement 2*). Wiskostatin (Wisko, 10 µM, Millipore Sigma) was used to perturb N-WASP activity in B-cells (*Peterson et al., 2004*), with DMSO used as a vehicle control. Splenic B-cells were pretreated with Wisko for 10 min at 37 °C before and during incubation with Fab'-PLB. The effectiveness of Wisko was determined by its inhibitory effects on the level of phosphorylated N-WASP in the B-cell contact zone using immunostaining and TIRF (*Figure 1—figure supplement 4A and C*). Possible effects of Wisko on WASP activation were evaluated by measuring the mean fluorescence intensity (MFI) of phosphorylated WASP in the contact zone of B-cells treated with or without Wisko for 10 min (*Figure 1—figure supplement 4B and D*). Blebbistatin (Bleb, 50 µM, Cayman Chemicals) was used to inhibit the NMII motor activity (*Seeley-Fallen et al., 2022*). Splenic B-cells were pretreated with Bleb for 20 min before and during incubation with Fab'-PLB.

### Analysis of the actin cytoskeleton

F-actin was visualized by phalloidin staining in fixed cells and by LifeAct-GFP or LifeAct-RFP expressed by primary B-cells from transgenic mice in live cells. Inner F-actin foci, visualized by both phalloidin staining and LifeAct-GFP, were identified using Fiji ImageJ based on the following three criteria: (1) ≥250 nm in diameter, (2) peak FI ≥twofold higher than the FI of a nearby area containing no foci, and (3) ≥1 µm away from the outer edge of the B-cell contact zone. The horizontal or vertical length of a focus was measured, and the lesser of the two values was used as its diameter. To analyze the spatio-temporal relationship between inner F-actin foci and the lamellipodial F-actin network, we generated 8 radially and equally spaced kymographs from each cell using time-lapse images and MATLAB [*Source code 9*]. Each kymograph was either classified as contracting or not contracting based on the movement of the leading edge of the B-cell contact zone. The percentage of eight kymographs from each cell exhibiting inner F-actin foci that could be traced back to lamellipodial F-actin when the B-cell switched from spreading to contraction was determined.

The relative speeds and lifetimes of F-actin foci were determined using three kymographs from each B-cell (positioned to track as many inner F-actin foci as possible) and F-actin foci emerging during a 60 s window right after B-cell maximal spreading. Only those remaining visible in individual kymographs for at least 4 s were analyzed. The time duration of each F-actin focus detected in a kymograph was used to determine the relative lifetime. The distances individual F-actin foci moved during their lifetimes were used to determine the relative speed. The term 'relative' has been used because the disappearance of an F-actin focus could be due to its movement away from the region used for the kymograph or vertically from the TIRF evanescent field of excitation.

To determine the rate of myosin recruitment, GFP-NMIIA MFI in the B-cell contact zone was plotted over time, and the slope for the initial segment of the MFI versus the time plot was determined using linear regression. The percentage of B-cells with the NMIIA ring was determined by visual inspection.

## Analysis of BCR-Fab' clusters

BCR clusters were identified by clustering of AF546-Fab' on PLBs using custom code by MATLAB [*Source code 10*]. We utilized the median FI of the Fab'-PLB within the same area outside but near the B-cell contact zone as the background. The median, but not the mean FI value, was used for background FI calculations to minimize fluctuations due to debris. To ensure that individual clusters with varying FI were detected as distinct objects, we used 16 graded thresholds from 1.1 to 4.1 fold of the background (0.2 fold apart) to acquire 16 sets of binary masks for each frame of time-lapse images from each cell. When clusters were detected by multiple thresholds at the same location, only the one identified by the highest threshold was retained. Objects that were smaller than 250 nm in diameter or could not be tracked for at least 20 s were eliminated. This allowed us to detect the position of BCR-Fab' clusters and track them over time until they merged with other clusters. The horizontal or vertical length of a focus was measured, and the lesser of the two values was used as its diameter. We chose the peak FI as a metric for the extent of Fab' clustering by the BCR, because it does not rely on the area occupied by each cluster. The rate of peak FI increase was determined by linear regression of peak FI versus time curves for each cluster for a given window of time.

## BCR signaling

Splenic B-cells were incubated with AF546-Fab'-PLB at 37 °C and fixed at 1, 3, 5, 7, and 9 min. After fixation, cells were stained for phosphorylated CD79a (pCD79a, Y182) (Cell Signaling Technology), Syk (pSyk, Y519/520) (Cell Signaling Technology), SHIP-1 (pSHIP-1, Y1020) (Cell Signaling Technology), or total Syk or SHIP-1 proteins (Thermo Fisher Scientific). We identified BCR-Fab' clusters as described. We determined the peak FI of AF546-Fab' and the MFI of AF546-Fab', pCD79a, pSyk, pSHIP-1, Syk, or SHIP-1 within each cluster. The ratio of pCD79a MFI to AF546-Fab' MFI in the same cluster was used to estimate the relative phosphorylation level of BCRs in that cluster. This MFI ratio was plotted against AF546-Fab' peak FI to analyze the relationship between BCR phosphorylation and BCR density of individual clusters. We calculated the MFI ratio of Syk to Fab' or SHIP-1 to Fab' in individual clusters to estimate the relative recruitment level of Syk or SHIP-1 to BCR clusters. We plotted these MFI ratios against Fab' peak FI to determine the relationship between Syk and SHIP-1 recruitment levels and BCR density in individual clusters. We calculated the MFI ratio of pSyk to Fab' or pSHIP-1 to Fab' to estimate the amount of pSyk or pSHIP-1 relative to BCR in individual clusters. We plotted these MFI ratios against Fab' peak FI to determine the relationship between Syk or SHIP-1 phosphorylation and BCR density in individual clusters. We identified pSyk or pSHIP-1 puncta using the criteria: ≥1.3 fold of the background staining outside the B-cell contact zone and ≥250 nm in diameter. We calculated the MFI ratio of pSyk to Fab' or pSHIP-1 to Fab' in individual puncta to estimate the amount of pSyk or pSHIP-1 relative to BCR in individual puncta. We plotted these MFI ratios against Fab' peak FI to determine the relationship between Syk or SHIP-1 phosphorylation and BCR density in individual puncta. We determined the fraction of clusters with graded increases in Fab' peak FI (every 20 FI units) out of the total to analyze the distribution of clusters with different Fab' peak FI. We also determine the average MFI ratios of pCD79a, Syk, pSyk, SHIP-1, and pSHIP-1 relative to Fab' in each graded cluster population. We further divided AF546-Fab' clusters into three populations based on their peak AF546-Fab' FI, relatively low (<190), medium (190-280), and high (>280, detected only in contracted cells) and compared the MFI ratios of pCD79a, pSyk, pSHIP-1, Syk, and SHIP-1 relatively to AF546-Fab' among the three populations using statistical analysis described below. The data were generated from three independent experiments with 20~23 (60~69 total) cells and 125~140 individual clusters (1500~1680 total) per condition, per time point, and per experiment.

## Statistical analysis

Statistical analysis was performed using the Mann-Whitney U non-parametric test for unpaired groups having different sample sizes or the Student's t-test for paired groups with the same sample size. To compare curves, the Kolmogorov-Smirnov test was used. Statistical analyses were performed in Microsoft Excel, GraphPad Prism, and MATLAB. All data are presented as mean ± SEM (standard error of the mean). When testing multiple hypotheses, p-values acquired using t-tests were corrected using the Benjamini-Hochberg/Yekutieli method for false discovery rate control.

## MATLAB scripts

All MATLAB scripts used for this study are available as Source code files in the supplemental materials.

## Acknowledgements

This work was supported by NIH grants GM064625 (NWA and WS), GM145313 (AU), and AI122205 (AU and WS). We would like to acknowledge Amy Beaven and Kenneth Class for technical support from the Confocal Microcopy and Flow Cytometry cores. We would like to thank Dr. Norma W Andrews (University of Maryland, College Park) for her careful reading and critical comments on the manuscript.

## Additional information

### Funding

| Funder | Grant reference number | Author |
| --- | --- | --- |
| National Institute of Allergy and Infectious Diseases | AI122205 | Arpita Upadhyaya |
| National Institute of General Medical Sciences | GM064625 | Wenxia Song |

The funders had no role in study design, data collection and interpretation, or the decision to submit the work for publication.

### Author contributions

Anshuman Bhanja, Data curation, Software, Formal analysis, Investigation, Visualization, Writing – original draft; Margaret K Seeley-Fallen, Data curation, Writing – review and editing; Michelle Lazzaro, Data curation; Arpita Upadhyaya, Supervision, Funding acquisition, Writing – review and editing; Wenxia Song, Conceptualization, Supervision, Funding acquisition, Writing – original draft, Project administration, Writing – review and editing

### Author ORCIDs

Wenxia Song [ID] http://orcid.org/0000-0001-8795-8657

### Ethics

All animals were handled according to approved institutional animal care and use committee (IACUC) protocols (R-JUL-22-34) of the University of Maryland, College Park.

Reviewer #1 (Public Review): https://doi.org/10.7554/eLife.87833.3.sa1
Reviewer #2 (Public Review): https://doi.org/10.7554/eLife.87833.3.sa2
Reviewer #3 (Public Review): https://doi.org/10.7554/eLife.87833.3.sa3
Author Response https://doi.org/10.7554/eLife.87833.3.sa4

## Additional files

### Supplementary files

• MDAR checklist

• Source code 1. Acquiring masks of B-cell contact zones from IRM images. This MATLAB script inputs raw 16-bit .nd2 images and outputs a binary mask of all detected contact zones for further analysis.

• Source code 2. Acquiring a mask of a B-cell contact zone from an IRM time-lapse image. This MATLAB script tracks the contact zone of a B-cell from time-lapse images and outputs a time-lapse mask file.

• Source code 3. Analyzing time-lapse images frame-by-frame for each B-cell. This MATLAB liveScript comprises several sections, including Source codes 5-8. It inputs time-lapse 16-bit .tiff images of a single cell consisting of IRM and two fluorescent channels. It outputs contact zone area

(μm$^2$), MFI, and TFI as an Excel file.

• Source code 4. Saving sets of time-lapse images to specified directory locations. This MATLAB script saves a single time-lapse file for further analysis.

• Source code 5. Quantifying fluorescence intensity from time-lapse images. This MATLAB script analyzes fluorescent intensity in time-lapse images of three-channels: IRM and two fluorescent channels.

• Source code 6. Exporting quantified data to an Excel file. This MATLAB script exports time-lapse data for one B-cell into a specified Excel file.

• Source code 7. Generating kymographs from a specified channel of a time-lapse TIRF image. This MATLAB script generates eight radially equally spaced kymographs from time-lapse images of a B-cell consisting of IRM and two fluorescent channels.

• Source code 8. Quantifying mean (MFI) and total fluorescence intensity (TFI) in the B-cell contact zone from TIRF images. This MATLAB script inputs a list of binary masks of B-cell contact zones, quantifies data using *Source code 3*, and generates a data table for each contact zone.

• Source code 9. Quantifying contact zone area, MFI, and TFI from one three-channel 16-bit .nd2 image. This MATLAB script inputs a three-channel image, including an IRM, and two fluorescent channels (labeled 'red' and 'green'), and a mask of all B-cell contact zones in the image. It calculates and outputs the contact zone area (μm$^2$) and MFI and TFI of the 'red' and the 'green' channels.

• Source code 10. Detecting fluorescent puncta from still or time-lapse TIRF images of B-cells. This MATLAB script identifies AF546-Fab' clusters and signaling molecule puncta from 16-bit .nd2 images consisting of three channels: IRM, AF546-Fab', and (stained) signaling molecule. Inputs B-cell contact zone masks, utilizes *Source code 11* to quantify AF546-Fab' peak intensity, Fab' MFI, and signaling molecule MFI in each detected cluster, and generates the data table.

• Source code 11. Quantitative analysis of individual AF546-Fab' clusters or signaling molecule puncta. This MATLAB script quantifies Fab' peak intensity, Fab' MFI, and signaling molecule MFI in each cluster or puncta detected in TIRF images.

• Source code 12. Quantitative analysis of AF546-Fab' cluster from time-lapse TIRF images. This MATLAB script is composed of several sections with some sub-functions (*Source code 13 and 14*). It inputs time-lapse masks delineating B-cell contact zones from time-lapse images, identifies AF546-Fab' clusters in one B-cell contact zone for every frame in the original time-lapse TIRF image, generates tracks connecting the detected clusters, analyzes the peak intensity and MFI of AF546-Fab' in each cluster, and generates a data table.

• Source code 13. Tracking AF546-Fab' clusters in time-lapse TIRF images. This MATLAB script tracks detected clusters by creating a frame-by-frame association between two consecutive frames of time-lapse TIRF images.

• Source code 14. Quantitative analysis of AF546-Fab' clusters tracked within time-lapse images. This MATLAB script analyzes detected AF546-Fab' clusters and quantifies AF546 peak intensity and MFI in that cluster.

### Data availability
All data generated or analyzed during this study are included in the manuscript.

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
