## [Editor Report · eLife assessment]

This is an **important** study highlighting a distinct role of WASP dependent actin foci in B cell antigen receptor signalling. The evidence supporting the conclusions is **compelling**. The proposal of higher molecular density in B cell receptor clustering leading to kinase exclusion and attenuated signalling is provocative as it contrasts with models for other antigen receptors.

---

## [Referee Report · Reviewer #1 (Public Review)]

In this study, the authors demonstrated a new model that B cell contraction after antigen encountering was dependent on N-WASP-branched actin polymerization. This statement is achieved by a systemic comparison of genetic modified mice vs wild type mice or inhibitor treated cells vs control cells. By imaging how B cells interact with antigen-coated planar lipid bilayer, the authors further suggested that the contraction event may provide B cells a channel to dismiss downstream kinase for a purpose to attenuate B cell activation signaling.

In this revised version, the authors have fully addressed my concerns raised against the initial submission of their studies.

---

## [Referee Report · Reviewer #2 (Public Review)]

Bhanja et al have examined how actin polymerization switch B-cell receptor (BCR) signaling from amplification to attenuation. The authors have examined B cell spreading and contraction using lipid bilayers to assess the molecular regulation of BCR signalling during the contraction phase. Their data provide evidence for that N-WASP activated Arp2/3 generates centripetally moving actin foci and contractile actomyosin from lamellipodia actin networks. This generates BCR dense foci that pushes out both stimulatory kinases and inhibitory phosphatases. The study provides novel insight into how B cells upon activation attenuate BCR signalling by contraction of the actin cytoskeleton and clustering of BCR foci and this dynamic response is mediated by N-WASP and Arp2/3.

Strengths: The manuscript is well written and results, methods, figures and legends described in detail making it easy to follow the experimental setup, analysis, and conclusions. The authors achieved their aims, and the results support their conclusions.

Weaknesses: Minor. The working hypothesis of molecular crowding as a way to push out signalling molecules from the BCR dense foci is interesting. The authors provide evidence for that this is an active process mediated by N-WASP - Arp2/3 induced actin foci. Another possibility discussed in the revised version is that BCR dense foci formation is an indirect consequence of lamellipodia retraction. Future works should define the specific role of N-WASP, Arp2/3 and actin in the process to form BCR dense foci, especially as the BCR continue to signal in the cytoplasm.

---

## [Referee Report · Reviewer #3 (Public Review)]

This work shows how, in the formation of the immune synapse, the B cell controls the contraction phase, the formation and retraction of actin structures concentrating the antigen (actin foci), and, ultimately, global signal attenuation. The authors use a combination of TIRF microscopy and original image quantification to show that Arp2/3 activated by N-WASP controls a pool of actin concentrated in foci (situated in the synapse), formed and transported centripetally towards the center of the synapse through myosin II mediated contractions. These contractions concentrate the B cell receptors (BCR) in the center, promote disassembly of the stimulatory kinase Syk as well as the the disassociation from the BCR of the inhibitory phosphatase SHIP, process which entails the attenuation of the BCR signal.

The author prove their claims by mean of thorough image analysis, mainly observing and quantifying the fluorescence and the dynamics of single clusters of antigen and actin foci and analyzing two-colors dynamical images. They perform their observation in control cells, on pharmacologically perturbed cells where the action of Arp2/3 or N-WASP is inhibited, and on modified primary cells (primary derived from genetically engineered mice) to silence N-WASP or WASP. The work is sound and complete, the experiments technically excellent and well explained.

In the reviewed manuscript the authors answer to all referees' suggestions and add new data and comments to the manuscript. In particular by suppressing NMII activation (with Blebbistatin), they show that NMII contraction plays a role (in coordination with N-WASP mediated actin polymerization) in the generation of actin foci ring-like structures.

This work adds an important information to the current view of B cell activation, in particular it links the contraction phase to the actin foci that have been recently characterized. Moreover, the late phase of the immune synapse formation is poorly investigated, but it is crucial for the fate of the cell: this work provides an explanation for the attenuation of the signal that might lead to the termination of the synapse.

---

## [Author Response]

The following is the authors’ response to the original reviews.

**Reviewer #1 (Public Review):**
The first major issue is related to the imaging and tracking experiment to examine the formation and migration of F-actin foci as illustrated in figure 3. The formation and centripetally migration of F-actin foci is a significant finding of this MS for the promotion of B cells to switch from spreading to contraction response. Thus, I may suggest to recommend the authors to conduct one more rigorous fluorescent molecular tracking experiment to confirm this phenomenon. Molecular tracking usually requires low labeling density, and the lifeact-GFP labeling here do not meet this requirement which may cause misidentification of the moving molecules. Permeable dye-based fluorescent speckle microscopy is recommended here to track the actin foci if applicable (P. Risteski, Nat. Rev. Mol. Cell Biol., 2023, DOI: 10.1038/s41580-023-00588w & K. Hu, et al, Science, 2007, 315, 111-115).

We thank the reviewer for the suggestion. We conducted the suggested experiment using membrane-permeable SiR-actin to track B-cell actin dynamics. Unfortunately, two significant issues prevented us from confirming the LifeAct-GFP results using fluorescent speckle microscopy. First, the concentration of SiR-actin required to visualize F-actin in the contact zone of mouse primary B-cells was relatively high due to their smaller sizes (~6 µm diameter) and non-adherent nature. With such a relatively high concentration of SiR-actin, we could not perform fluorescent speckle microscopy. Second, we observed that SiR-actin appeared to stabilize actin structures and reduce actin dynamics, further limiting its use in studying actin dynamics in B-cells.

Additionally, kymograph is used for foci tracking in figure3 and figure4. Kymograph is indeed a powerful tool for tracking cell protrusion and retraction but is not fairly suitable here, since a Factin focus is a concentrated point which may not move strictly along the selected eight lines generating kymograph. Other imaging processing method should be used to track the foci, for example, time series max projection is recommended if applicable.

We thank the reviewer for the suggestion and have tried the time series max projection. Unfortunately, it did not provide the resolution to identify individual actin foci, again probably due to the small size of primary mouse B-cells. While kymographs may not track the entire paths of these moving foci, we believe that the conclusions drawn from the kymography analysis in Figure 3 and 4 are reasonable. We generated eight kymographs for each cell in Figure 3 and three kymographs for each cell in Figure 4 to follow as many actin foci as possible within the spreading to contraction transition time window. Our analysis in Figure 3 identifies the fraction of actin foci originating from lamellipodia. In Figure 4, we used the kymographs to trace the path of putative clusters and used these to calculate their relative lifetimes and speed. While this is not what was suggested by the reviewer, our analysis provides qualitatively similar information to the time series max projection and reasonable comparisons between contracted and noncontracted cells, inhibitor-treated and untreated cells, and wild-type and WASP KO cells.

The second major issue is about the relationship between actin foci formation and NMII recruitment in figure 5. The author concludes that 'N-WASP and Arp2/3 mediated branched actin polymerization promotes the recruitment and the reorganization of NMII ring-like structures by generating inner F-actin foci in the contact zone'. However, there is a lack of strong evidence to directly show the mechanism by which myosin is recruited and the up and down stream relationship between actin foci migration and myosin recruitment. Since myosin-induced actin retrograde flow is a classical model in adherent cells, is it possible that, here also in activated B cells, the recruited myosin driven the formation and migration of actin foci? This reviewer may recommend the author to investigate whether Myosin blocking (e.g., using Y27632) can eliminate the F-actin foci formation and migration.

This is an excellent suggestion! In the revised manuscript, we have included new data showing that treatment with the non-muscle myosin II motor inhibitor blebbistatin, which is known to inhibit B-cell contraction but not spreading on Fab’-PLB (Seeley-Fallen et al. 2022. Frontiers in Immunology), interferes with the formation of inner actin foci ring-like structures, which are associated with B-cell contraction. These results together suggest that the generation of inner actin foci ring-like structure depends on the coordination between N-WASP-mediated actin polymerization and myosin contractile activity. We chose to use blebbistatin rather than Y27632 to inhibit non-muscle myosin II because in addition to the ROCK pathway, myosin light chain kinase can also activate myosin II, and Y27632 may have additional effects besides inhibiting myosin activity. The new data are shown in Figure 5G and H and discussed in the revised manuscript.

**Reviewer #2 (Public Review):**
Weaknesses: Minor as listed below. The working hypothesis of molecular crowding as a way to push out signalling molecules from the BCR dense foci is interesting. The authors provide evidence for that this is an active process mediated by N-WASP - Arp2/3 induced actin foci. Another possibility is that BCR dense foci formation is an indirect consequence of lamellipodia retraction. Future works should define the specific role of N-WASP, Arp2/3 and actin in the process to form BCR dense foci, especially as the BCR continue to signal in the cytoplasm.

We thank the reviewer for the comments. We have included the possibility that lamellipodial retraction may be involved in increasing the molecular density of BCR clusters and suggested future studies on the potential roles of N-WASP-dependent inner actin foci and actomyosin structures in BCR internalization and intracellular signaling in the Discussion section.

**Reviewer #3 (Public Review):**
The author prove their claims by mean of thorough image analysis, mainly observing and quantifying the fluorescence and the dynamics of single clusters of antigen and actin foci and analyzing two-colors dynamical images. They perform their observation in control cells, on pharmacologically perturbed cells where the action of Arp2/3 or N-WASP is inhibited, and on modified primary cells (primary derived from genetically engineered mice) to silence N-WASP or WASP. The work is sound and complete, the experiments technically excellent and well explained. Some experiments and discussions are objectively harder to describe, and given the length of the work, the reader might find itself lost some times. A graphical abstract/summary of the main way N-WASP ultimately control signal attenuation would solve this minor point.

We greatly appreciate the reviewer’s confirmation of our data quality and are delighted to accept the reviewer’s suggestion. In the revised manuscript, we have included a new figure (Figure 10) in the Discussion section, summarizing the results presented in the manuscript as a working model.

**Reviewer #1 (Recommendations For The Authors):**
Some minor points:Figure 1C, E, G and I shows three individual symbols, indicating three independent experiments described in legend. Please double check for accuracy.It is better to show statistical data with representative repeat, not the merged means of independent experiments. For example, figure 1C even indicates three "0" data in CK-666 treated cells, meaning no contracting cell was found in ~75 cells, while there are other repeats showing 45% - 50% contracting cells. This applies to all figures involving individual cell imaging data, such as figure 2D, in which 30 cells from three independent experiments were pooled. The authors shall clearly state that those independent experiments are statistically indistinguishable before pooling the data.

We agree with the reviewer’s comments that these data have variability from individual mice, the quality of isolated primary B-cells, and the lateral mobility of planar lipid bilayers. To show the variability, we displayed the data from each experiment as individual data points. In the revised manuscript, we have utilized three colors of dots to represent three independent experiments in Figure 1C, E, G, and I, Figure 2B-G, and new Figure 5H, which show that the data from the three experiments have the same trend despite the variability.

In figure 7B-C, figure 8 and figure 9. The significant test results were hard to understand in which groups they compared. Please describe it in more detail in the figure legend or the method section.

In the legend, the authors claimed blue points in Figure 7B represented individual pCD79a clusters within an equal number of BCR clusters from each time points. The authors used means to qualify the change of blue points distribution. These shall be clearly stated in the Methods. Total BCR cluster numbers shall be shown also. This applies to Figure 7B, 7C, 7D and all figures in figure 8 and figure 9.

We thank the reviewer for pointing it out. We have revised Figures 7-9, where we utilized square braces to indicate groups of clusters (blue points) being compared. We have also provided additional information in the figure legend and Method sections.

**Reviewer #2 (Recommendations For The Authors):**
199-200: What is the consequence of increased WASP activation in N-WASP knockout B cells? Is this evaluated as increased pWASP activity and/or increased actin polymerization of WASP knockout B cells. Does WASP and N-WASP have an additive or counteractive effect on each other during spreading and contraction?

Indeed, the relationship between WASP and N-WASP, which are co-expressed in B-cells and other immune cells, is fascinating. Our previous studies, using WASP germline knockout, B-cellspecific N-WASP knockout, WASP and N-WASP double knockout mice, showed that WASP and N-WASP have both additive and counteractive effects during B-cell spreading, but B-cell contraction only depends on N-WASP (Liu et al. 2013. PLoS Biol). Double knockout B-cells fail to spread, and WASP knockout B-cells show reduced spreading but still contract, showing their additive effects. However, WASP and N-WASP suppress each other for activation, as detected by their phosphorylation. Phosphorylated WASP increases in the B-cell contact zone first, and phosphorylated N-WASP increases later when the phosphorylated WASP level decreases. Knocking out one of them enhances the phosphorylation of the other. Consequently, N-WASP knockout B-cells show increased spreading, probably due to enhanced activation of WASP, but exhibit delayed contraction. The revised manuscript has expanded the discussion on this area to relate it to the results presented in this manuscript.

560-563: Was Syk and SHIP-1 measured in the same cell? If not, the conclusion should be tempered.

Unfortunately, antibodies specific for Syk and SHIP-1 were from the same host, which did not allow us to stain them in the same cells. The revised manuscript has discussed this as a shortcoming of our work.

1204-1205: Explain better "three randomly positioned kymographs were generated" - how were they selected?

We apologize for this unclear sentence. The three kymographs were positioned to track as many inner F-actin foci as possible.

328: Change "abolished" to "reduced" to describe the data.354-356: Unclear sentence, please edit.1171: (H) should be (G).1325: "PI" should be "FI".

We thank the reviewer for finding these typos and unclear sentences. We have made the corrections accordingly.

Methods: The description of the TIRF microscopy method is good. Regarding the image analysis, it is somehow difficult to have a good understanding of what was analyzed just by reading the text. Please show an example of the pipeline for the analysis from a raw image and the processing steps.

Figure 6-figure supplement 2 shows the image analysis process for tracking Fab’ clusters. We utilized the same approach for the image analysis of Figures 7-9.

Discussion: Add a paragraph to state the limitations of the study. How do the findings here translate into in vivo activation of B cells and how can this be addressed based on the data presented in this study.

We thank the reviewer for the suggestion. In several paragraphs of the revised Discussion section, we have brought up the limitations of the study and how these limitations affect the data interpretation. In addition, we have added Figure 10 and the associated text to present our working model, which explains how our findings reveal the cellular mechanism by which BCR surface signaling amplification transitions into attenuation, likely occurring in vivo.

Figure 2: Add an example of the image analysis for foci determination. From the images, it is not always clear what is a foci and what is not which makes the "number of foci" data difficult to evaluate.

We have added arrows to Figure 2A to indicate all identified inner F-actin foci in images.

Figure 3: add a kymograph for the WKO analysis.

In the revised Figure 4, we have provided a kymograph of a WKO B cell.

Figure 4M: the analysis of the "relative speed" of the "WT" samples is lower compared to the other control samples "DMSO" and "CK-689". The conclusion is that WKO have similar "relative speed" as "WT" cells, but in fact the "WT" cells may have responded poorly in this experiment. What is the author's experience and explanation?

We agree that the relative speeds of inner actin foci in the contact zone of WT and WKO B-cells are relatively low compared to DMSO and CK-689. Based on our experience, this parameter is very sensitive to the lateral mobility of planar lipid bilayers. We could only perform one pair of conditions using live cell images each time. The WT and WKO experiments were done at the end and might use relatively aged liposomes. However, it did not affect the number of inner actin foci formed and their relative lifetime, consistent with their similar relative speeds. Unfortunately, we lost the LifeAct-GFP-expressing WKO mouse colony and cannot redo this experiment using freshly made liposomes within a reasonable time.

Figure 7B-D: Add a more detailed legend for the black and brown lines in the dot plots.

We have expanded the legend for Figure 7B-D to provide additional details.

Figure 8-9: Show representative images for SYK, pSYK, SHIP-1 and pSHIP-1. Add a more detailed legend for the black and brown lines in the dot plots.

We have provided representative images for Syk, pSyk, SHIP-1, and pSHIP-1 in revised Figure 8 and 9.

**Reviewer #3 (Recommendations For The Authors):**
From the paper one understands that NMII is recruited by the actin foci and this recruitment pushes the foci towards the center of the synapse, in what resembles a positive feedback. Could the authors better elucidate this point? What happen at the peak of NMII recruitment? Could this be a mechanism used by the cell to end the contact and detach (which probably cannot be observed in this experimental setup)?

This is an excellent comment! We have recently shown that NMIIA recruitment peaks right before B-cell contraction occurs, and inhibition of NMII by inhibitors or B-cell conditional knockout blocks B-cell contraction and enhances signaling (Seeley-Fallen et al. 2022. Frontiers in Immunology). In the revised manuscript, we have included new data showing that treatment with the NMII motor inhibitor blebbistatin, which is known to inhibit B-cell contraction but not spreading on Fab’-PLB (Seeley-Fallen et al. 2022. Frontiers in Immunology), interferes with the formation of inner actin foci associated with B-cell contraction. These results together suggest that the generation of inner actin foci depends on the coordination between N-WASP-activated actin polymerization and myosin contractile activity, supporting the reviewer’s comment. The new data are shown in Figure 5G and H and discussed in the revised manuscript.

Whether the recruited NMII pulls B-cells away from antigen-presenting surfaces remains an interesting question. We have previously shown that high-affinity interaction of surface BCRswith membrane-anchored antigen can cause NMII-dependent B-cell membrane permeabilization, which triggers lysosome exocytosis and lysosomal enzyme-mediated antigen cleavage, allowing antigen internalization and presentation to T-cells (Maeda et al. 2021. eLife). Furthermore, NMII is required for B cells to internalize surface antigens (Natkanski et al. 2013. Science). These results support the possibility that actomyosin structures formed during B-cell contraction may further drive B-cells to internalize antigen. We have discussed this interesting point in the revised manuscript.

Some experiments/quantification are a bit more complex than others and a reader might find hard to follow them (in particular figs 7,8 and 9). The comprehension could be improved by providing a guide to read them. E.g. it is not clear what the population distribution represents and it is not particularly affected by any manipulation. How were the group for test chosen? It seems they are based on intensity categories taken every 100 units: is it the case? even if arbitrary, this should be stated it in the legend.

We thank the reviewer for understanding the complexity of image analysis and pointing out the unclear points. Based on the reviewer’s comments, we have revised Figures 7-9 and the figure legend. We utilized square brackets to indicate groups of clusters (blue points) being compared. The comparison groups were chosen arbitrarily based on Fab’ peak fluorescence intensity every 90 units for Figure 7 and 8 and every 100 units for Figure 9.

Can the author speculate on how the actin organization passes from actin foci to recruitment of NMII and arc formation? Is it a rearrangement of the actin network (percolation) or simply recruitment of monomers?

Our previous and new results show that both N-WASP-activated Arp2/3 and NMII are required to form inner F-actin foci. Based on these results, we speculate that N-WASP and Arp2/3mediated actin polymerization may initiate the process and recruit NMII, and recruited NMII coordinates with actin polymerization to reorganize actin structures, promoting inner actin foci maturation and arc formation. We have included these possibilities in the revised discussion.

The role of SHIP recruitment as way to inhibit the signal downstream of the BCR is an interesting finding. Is this related to the termination of the synapse? Could we relate the time scales (accurately measured in this work) to contact times observed in vivo?

The reviewer raises an interesting question. In the discussion section, we have speculated that the actomyosin structures responsible for B-cell contraction are potentially the precursor cytoskeleton structures for antigen internalization. However, the relationship of B-cell contraction and signaling attenuation with the termination of the synapse remains unclear.

The BCR has been shown to be internalised mechanically: do these new data suggest a mechanisms for force generation in antigen internalization at the actin foci? Related to that, how do the dynamics of N-WASP recruitment relate to the force measurement highlighted in TractionForce Microscopy experiments (see for example Wang Sci.Signal. 2018, Kumari Nat.Comm.2019)? What happens in situation when the actin foci are unable to get transported, e.g. as on the more classical antigen on coverslip configuration?

Indeed, our results allow us to speculate that the actomyosin structures responsible for B-cell contraction potentially contribute to antigen internalization by mechanical forces. We previously showed that the B-cell-specific N-WASP knockout drastically reduced BCR internalization of soluble antigen (Liu et al. 2013. PLoS Biol), and that NMII is required for BCR internalization of membrane-associated antigen (Maeda et al. 2021. eLife and Natkanski et al. 2013. Science). The effect of N-WASP knockout on the internalization of membrane-associated antigen and traction forces generated at the contact membrane and whether traction forces are generated from the inner F-actin foci have not been determined but will be pursued in the future.

Our previous publication compared the BCR and actin dynamics of B-cells interacting with Fab’ tethered to planer lipid bilayers (Fab’-PLB) and cover glass (Fab’-G) (Ketchum et al. 2014. Biophys J). B-cells interacting with Fab’-G do not contract and generate inner F-actin foci and exhibit less dynamic BCR clusters and actin cytoskeleton than B-cells interacting with Fab’-PLB. Actin foci remain coincident with Fab’ clusters on glass rather than being positioned behind Fab’ clusters on PLB, thus driving their centripetal movement.

Minor remarks:When several experiments (mice) are presented in dot plots (e.g. fig 2D-G 4J-M), color dot plot (so called "smart plot") where each experiment is identified by a color, could be used to highlight the sample-to-sample variability.

This is an excellent suggestion. In the revised manuscript, we have utilized three shades of dots to represent the data points from three independent experiments.

Fig 6A: the fluorophore should be indicated in the picture (Fab'-AF546)

The suggested correction has been made.

Fig 6D: how is the contraction phase (purple rectangle) determined? Curve by curve or on the average curve? Please specify this in the legend.

The contraction phase (purple rectangle) was determined using the average curve of the contact area by IRM over time. We have added this sentence to the revised figure legend.

Minor typos in the material and methods: in some case C56BL/6 is written instead of C57BL/6Corrected.